# Sensory experience inversely regulates feedforward and feedback excitation-inhibition ratio in rodent visual cortex

**Nathaniel J Miska[1], Leonidas MA Richter[2,3], Brian A Cary[1], Julijana Gjorgjieva[2,3], Gina G Turrigiano[1]\***

[1]Department of Biology, Brandeis University, Waltham, United States; [2]Max Planck Institute for Brain Research, Frankfurt, Germany; [3]School of Life Sciences, Technical University of Munich, Freising, Germany

**Abstract** Brief (2-3d) monocular deprivation (MD) during the critical period induces a profound loss of responsiveness within binocular (V1b) and monocular (V1m) regions of rodent primary visual cortex. This has largely been ascribed to long-term depression (LTD) at thalamocortical synapses, while a contribution from intracortical inhibition has been controversial. Here we used optogenetics to isolate and measure feedforward thalamocortical and feedback intracortical excitation-inhibition (E-I) ratios following brief MD. Despite depression at thalamocortical synapses, thalamocortical E-I ratio was unaffected in V1b and shifted toward excitation in V1m, indicating that thalamocortical excitation was not effectively reduced. In contrast, feedback intracortical E-I ratio was shifted toward inhibition in V1m, and a computational model demonstrated that these opposing shifts produced an overall suppression of layer 4 excitability. Thus, feedforward and feedback E-I ratios can be independently tuned by visual experience, and enhanced feedback inhibition is the primary driving force behind loss of visual responsiveness.
DOI: https://doi.org/10.7554/eLife.38846.001

**\*For correspondence:**
turrigiano@brandeis.edu

**Competing interests:** The authors declare that no competing interests exist.

## Introduction

The fine-tuning of microcircuit function in primary sensory cortex requires sensory experience during an early critical period (*Espinosa and Stryker, 2012*; *Gainey and Feldman, 2017*; *Hensch, 2005*; *Hubel and Wiesel, 1970*), but the plasticity mechanisms that drive this refinement are not fully defined. In the visual system of rats and mice, the critical period extends from roughly postnatal days 21–35 (*Espinosa and Stryker, 2012*), when visual deprivation has catastrophic effects on visual function, including loss of visual responsiveness to the deprived eye (*Heynen et al., 2003*; *Frenkel and Bear, 2004*), reduced visual acuity (*Fagiolini et al., 1994*), loss of tuning to many stimulus characteristics (*Espinosa and Stryker, 2012*), and a suppression of firing within primary visual cortex (V1) (*Hengen et al., 2013*; *Hengen et al., 2016*). Even very brief (1–3 days) monocular deprivation (MD) during the critical period induces a rapid loss of visual responsiveness to the deprived eye in both binocular (V1b, *Heynen et al., 2003*; *Frenkel and Bear, 2004*) and monocular (V1m; *Heynen et al., 2003*; *Kaneko et al., 2008*; *Wang et al., 2011*) regions of V1. This rapid loss of deprived-eye responsiveness is thought to be primarily cortical in origin (*Espinosa and Stryker, 2012*; *Gainey and Feldman, 2017*; *Sommeijer et al., 2017*; *Wang et al., 2011*), but it remains controversial which forms of intracortical plasticity underlie it. In particular, is it unclear whether classic Hebbian long-term depression (LTD) is the sole mediator (*Heynen et al., 2003*; *Smith et al., 2009*), or whether more complex changes in cortical circuitry that modify the relative recruitment of excitation and inhibition are equally or perhaps more important (*House et al., 2011*; *Kuhlman et al., 2013*; *Li et al., 2014*). Here we address this question by using optogenetic, electrophysiological,

and modeling approaches to quantify the impact of brief MD on excitation-inhibition balance within defined feedforward and feedback pathways in V1.

Traditionally, the loss of visual responsiveness after brief MD has been ascribed to LTD at excitatory synapses onto pyramidal neurons (*Crozier et al., 2007*; *Espinosa and Stryker, 2012*; *Frenkel and Bear, 2004*; *Heynen et al., 2003*; *Lambo and Turrigiano, 2013*; *Smith et al., 2009*), including LTD at thalamocortical synapses onto excitatory neurons in layer 4 (*Crozier et al., 2007*; *Khibnik et al., 2010*; *Wang et al., 2013*). In this view, loss of responsiveness in layer 4 is a simple reflection of reduced efficacy at thalamocortical synapses onto excitatory postsynaptic neurons. In contrast, a number of recent studies have demonstrated that cortical inhibitory circuitry is also plastic and can be modified by brief MD (*Kannan et al., 2016*; *Kuhlman et al., 2013*; *Maffei et al., 2006*; *Nahmani and Turrigiano, 2014*; *Sun et al., 2016*). Given that the relationship between excitation (E) and inhibition (I) is critically important for neocortical information processing (*Isaacson and Scanziani, 2011*; *Haider and McCormick, 2009*), E-I ratio is likely a more relevant measure of circuit excitability than excitation alone (*House et al., 2011*; *Li et al., 2014*; *Xue et al., 2014*). However, it is currently unknown whether brief MD alters E-I ratio within specific microcircuit motifs in V1.

Primary sensory neocortical regions such as V1 have a stereotyped microcircuit architecture (*Douglas et al., 1989*; *Van Hooser, 2007*). Sensory information is relayed to cortex via thalamocortical inputs, which excite both excitatory pyramidal neurons and inhibitory interneurons within layer 4, forming a 'feedforward' circuit. In addition, pyramidal neurons recurrently excite other pyramidal neurons and interneurons within layer 4, forming an intracortical 'feedback' circuit. While these two circuits are intimately related and share both excitatory and inhibitory elements, they ultimately provide separable contributions to sensory-evoked cortical activity (*Reinhold et al., 2015*). Furthermore, feedback recurrent inputs greatly outnumber feedforward thalamocortical inputs (*Ahmed et al., 1994*; *Bopp et al., 2017*; *Douglas et al., 1989*), suggesting that the layer 4 feedback circuit is poised to disproportionately affect network excitability. The extent to which the feedforward thalamocortical and feedback intracortical components of synaptic drive may be independently tuned by experience-dependent plasticity remains an unexplored question.

To characterize changes in thalamocortical versus intracortical circuits following brief MD, we performed MD for 2–3 days then used optogenetic methods to isolate and probe these respective circuit elements in layer 4 of V1b and V1m. First, we expressed channelrhodopsin-2 (ChR2) in thalamocortical afferents in layer 4 and recorded from layer 4 pyramidal neurons after brief MD. Surprisingly, although thalamocortical inputs were indeed depressed by brief MD as expected, the E-I ratio was either unchanged (V1b) or shifted toward excitation rather than inhibition (V1m); this latter effect was mediated by a disproportionate depression of thalamocortical synaptic strength onto layer 4 PV+ interneurons compared to neighboring pyramidal neurons. In contrast, probing intracortical E-I ratio in V1m through sparse expression of ChR2 in layer 4 pyramidal neurons revealed that this local feedback circuit was significantly shifted toward inhibition following brief MD. We modeled these opposite shifts in feedforward versus feedback E-I ratio and found that they produced an overall depression in network activation over a wide range of parameters. These data suggest that contrary to the prevailing view, brief MD does not suppress feedforward thalamocortical drive onto layer 4 pyramidal neurons, and the loss of responsiveness within layer 4 arises instead through reduced intracortical amplification by the local feedback circuit.

## Results

In rats and mice, brief (1–3 d) MD during the critical period reduces visual responsiveness in both V1m (*Hengen et al., 2013*; *Hengen et al., 2016*; *Heynen et al., 2003*; *Kaneko et al., 2008*) and V1b (*Heynen et al., 2003*; *Frenkel and Bear, 2004*; *Kaneko et al., 2008*; *Khibnik et al., 2010*). The extent to which this reflects thalamocortical LTD (*Crozier et al., 2007*; *Smith et al., 2009*) versus reorganization of E-I networks (*Kuhlman et al., 2013*; *Maffei et al., 2006*) remains controversial. Here we use optogenetics to probe and compare the relative strengths of excitation and inhibition in thalamocortical feedforward and intracortical feedback circuits within layer 4 of V1. Except where noted, recordings were targeted to V1m, which allowed us to obtain data from deprived and control hemispheres of the same animals, and removed ambiguity about the eye-specific source of drive. Experiments were performed on critical-period Long-Evans rats (*Figure 1—figure supplement 1*, *Figure 2* and *Figure 3*), or C57BL/6J mouse lines when labeling of specific cell types was required

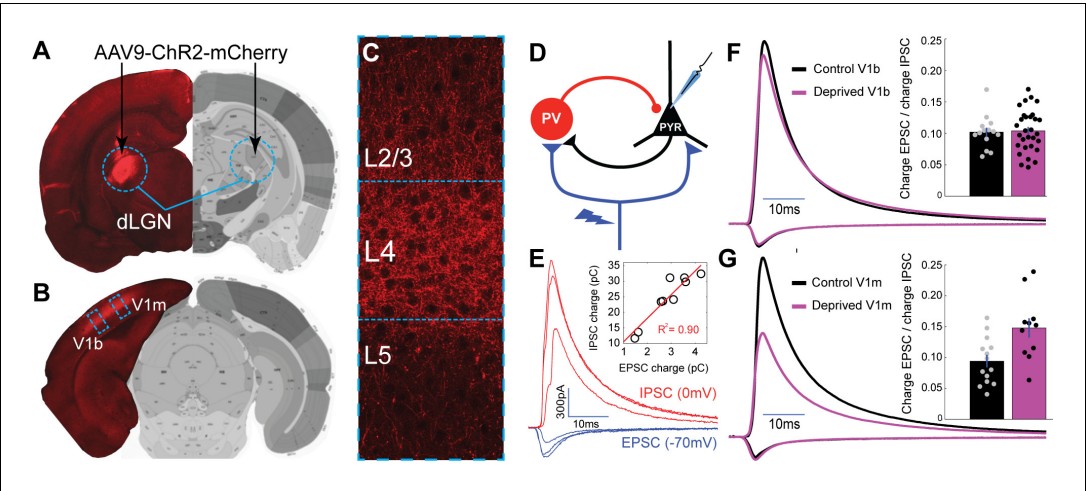

**Figure 1.** Brief MD increases thalamocortical-evoked Excitation-Inhibition (E–I) ratio in V1m. (**A**) Stereotaxic injection of AAV-ChR2-mCherry into dLGN (red) (**B**) leads to reporter expression within thalamocortical axons in V1m and V1b, (**C**) with densest innervation in layer 4 (**L4**). (**D**) Layer 4 pyramidal neurons were voltage-clamped in normal ACSF while stimulating local ChR2-mCherry+ thalamocortical axons with 473 nm light. (**E**) Representative example traces from a single neuron): brief (2 ms) stimuli were given at increasing laser intensity while alternating between the experimentally determined inhibitory and excitatory reversal potentials (−70 mV and 0 mV, respectively) to record paired monosynaptic thalamocortical-evoked EPSCs and disynaptic thalamocortical-evoked IPSCs. Inset: EPSC charge plotted versus IPSC charge for individual E-I pairs from this example neuron, with linear fit plotted in red ($R^2 = 0.90$). (**F**) Mean EPSC charge-normalized EPSC and IPSC traces for control (black) and deprived (magenta) neurons within V1b. Inset: Mean E-I charge ratio for control (black) and deprived (magenta) neurons (control n = 14 neurons, deprived n = 30 neurons, from six animals; p = 0.80, 2-sample t-test). (**G**) Same as F), but for neurons within V1m. Inset: Mean E-I charge ratio for control (black) and deprived (magenta) neurons (control n = 14 neurons, deprived n = 11 neurons, from seven animals; p = 0.0063, 2-sample t-test). Atlas images in A) and B) adapted from Allen Mouse Brain Atlas (***Dong, 2008***).

DOI: https://doi.org/10.7554/eLife.38846.002

The following source data and figure supplement are available for figure 1:

**Source data 1.** Source Data for *Figure 1*.
DOI: https://doi.org/10.7554/eLife.38846.004
**Figure supplement 1.** Measuring thalamocortical E-I ratio.
DOI: https://doi.org/10.7554/eLife.38846.003

(***Figure 1***, ***Figure 3***, ***Figure 4***, ***Figure 5***, ***Figure 6***, ***Figure 7*** and ***Figure 8***). MD was performed for two or four days to capture the period of maximum deprived-eye response suppression (***Frenkel and Bear, 2004***; ***Kaneko et al., 2008***; ***Hengen et al., 2016***); these data were combined, as changes in E-I ratio were similar for both lengths of deprivation.

## Thalamocortical-evoked E-I ratio following brief MD

Thalamocortical inputs to layer 4 target GABAergic interneurons as well as pyramidal neurons, and thalamocortical activation evokes a mix of monosynaptic excitation and disynaptic inhibition onto layer 4 pyramidal neurons (***Cruikshank et al., 2010***; ***Kloc and Maffei, 2014***). The ratio of evoked excitation and inhibition (E-I ratio) has been shown to exhibit significantly less variability than excitation or inhibition alone (***Xue et al., 2014***), suggesting it represents a conserved measure of excitability regulated at the level of the neuron or local microcircuit. If, as previously suggested (***Crozier et al., 2007***; ***Espinosa and Stryker, 2012***; ***Khibnik et al., 2010***; ***Smith et al., 2009***), LTD at thalamocortical to pyramidal synapses is the major driver of the loss of visual responsiveness in layer 4, then the ratio of thalamocortical-evoked excitation to thalamocortical-evoked inhibition should shift to favor inhibition after brief MD, but this prediction has never been tested.

We selectively labeled primary thalamocortical afferents in V1 with a ChR2-mCherry fusion protein by stereotaxically injecting adeno-associated virus (AAV-ChR2-mCherry) bilaterally into the dorsal lateral geniculate nucleus (dLGN) of the thalamus (***Figure 1A***, ***Kloc and Maffei, 2014***; ***Wang et al.,***

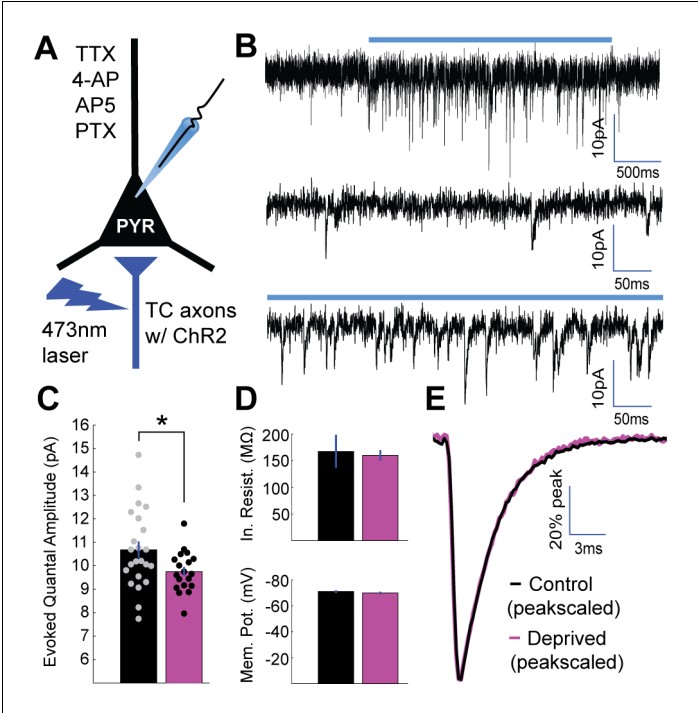

**Figure 2.** Brief MD reduces thalamocortical quantal amplitude onto layer 4 pyramidal neurons. (A) Whole-cell recordings were obtained from layer 4 pyramidal neurons in V1m in a drug cocktail of TTX, 4-AP, AP5, and PTX to isolate excitatory quantal events while stimulating local ChR2-mCherry+ thalamocortical (TC) axons (red) with 473 nm light. (B) Top: example recording of quantal events during 2 s laser stimulation (blue bar). Middle: Pre-stimulus spontaneous mEPSCs. Bottom: Evoked quantal events during laser stimulation. (C) Mean evoked quantal amplitudes for control (black) and deprived (magenta) neurons (control n = 23 neurons deprived n = 19 neurons, from 15 animals; p = 0.033, 2-sample t-test). (D) Mean input resistance (top) and resting membrane potential (bottom) for control and deprived neurons. (E) Overlaid peak-scaled evoked event waveform averages for control (black) and deprived (magenta) neurons, to illustrate kinetics. For all bar plots here and below, circles represent individual values, and error bars indicate ±SEM.

DOI: https://doi.org/10.7554/eLife.38846.005

The following source data and figure supplement are available for figure 2:

**Source data 1.** Source Data for *Figure 2*.
DOI: https://doi.org/10.7554/eLife.38846.007
**Figure supplement 1.** Evoked thalamocortical quantal events and spontaneous mEPSCs show similar amplitude distributions and event kinetics.
DOI: https://doi.org/10.7554/eLife.38846.006

*2013*). Following a 1.5–2 wk period to allow expression and transport of ChR2, we observed dense mCherry-positive thalamocortical axon terminals in V1, with strongest expression in layer 4, as expected (*Figure 1B,C*). To measure feedforward thalamocortical E-I ratio, we used brief (2 ms) pulses of blue laser light to activate thalamocortical afferents while holding postsynaptic layer 4 pyramidal neurons at the experimentally-determined reversal potential for excitation or inhibition (*Figure 1D,E*). Paired monosynaptic EPSCs versus disynaptic IPSCs scaled linearly at different stimulus intensities (*Figure 1E*, inset), indicating that thalamocortical E-I ratio is conserved as additional thalamocortical axons are recruited.

In contrast to the prediction of the thalamocortical LTD hypothesis, in V1b (where thalamocortical input arises from both the deprived and non-deprived eyes) E-I ratio did not shift to favor inhibition following brief MD, but instead shifted slightly but not significantly in the opposite direction (*Figure 1F*, control E-I ratio = 0.106 ± 0.007, deprived E-I ratio = 0.108 ± 0.006, p = 0.80). Even more strikingly, in V1m (where all thalamocortical input is driven by the contralateral eye) we found that thalamocortical-evoked E-I ratio shifted significantly to favor excitation in deprived neurons in

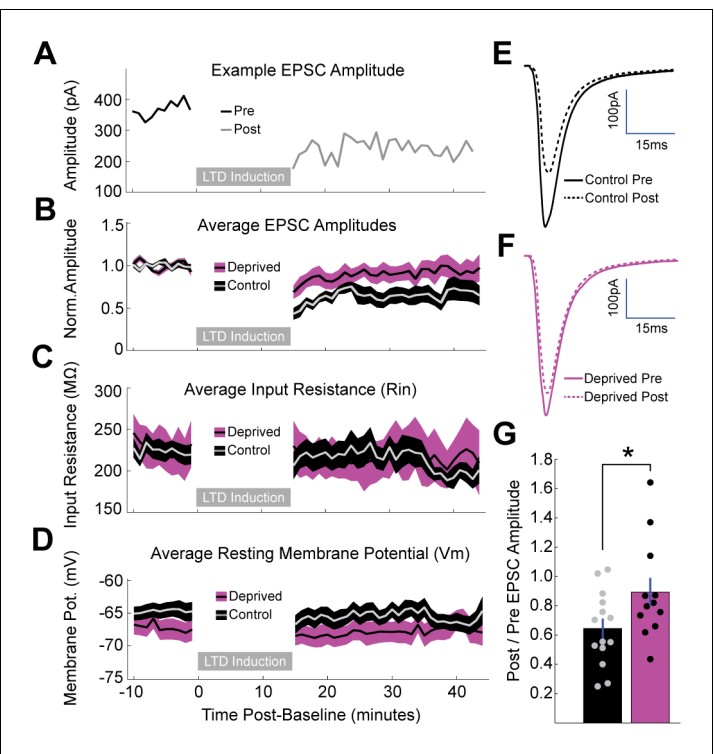

**Figure 3.** Brief MD occludes induction of LTD at thalamocortical synapses onto layer 4 pyramidal neurons. (**A**) Representative example of LTD induction from a nondeprived neuron in V1m, with pre-induction evoked EPSC amplitudes plotted in black, post-induction amplitudes plotted in grey, and LTD induction period represented as a grey bar. (**B**) Mean, baseline-normalized evoked EPSCs plotted for control (grey with black shading) and deprived (black with magenta shading) neurons. Shaded region represents SEM. (**C**) Mean input resistance (Rin) and (**D**) mean resting membrane potential (Vm) plotted for control and deprived neurons. (**E, F**) Average EPSC waveforms for control neurons (**E**) pre-induction (black, solid) and post-induction (black, dashed); and deprived neurons (**F**) pre-induction (magenta, solid) and post-induction (magenta, dashed). (**G**) Mean post/pre EPSC amplitude ratio for control (black) and deprived (magenta) neurons (control n = 14 neurons, deprived n = 12 neurons, from eight animals; p = 0.042, 2-sample t-test).

DOI: https://doi.org/10.7554/eLife.38846.008

The following source data is available for figure 3:

**Source data 1.** Source Data for *Figure 3*.

DOI: https://doi.org/10.7554/eLife.38846.009

---

both C57BL/6J mice (*Figure 1G*, control E-I ratio = 0.094 ± 0.010, deprived E-I ratio = 0.148 ± 0.016, p < 0.01) as well as in Long-Evans rats (*Figure 1—figure supplement 1A*). There was no change in passive properties between deprived and control neurons (*Figure 1—figure supplement 1B*).

To verify that evoked thalamocortical EPSCs were monosynaptic with this stimulation paradigm, we perfused TTX+4-AP and assessed EPSCs again (*Cruikshank et al., 2010*; *Ji et al., 2016*; *Petreanu et al., 2009*). At the stimulus intensities used, evoked thalamocortical EPSC kinetics and peak latency were unchanged following perfusion of TTX+4-AP (*Figure 1—figure supplement 1C*, control peak latency = 10.5 ± 0.27 ms, TTX+4-AP peak latency = 10.6 ± 0.28 ms, p = 0.82), while IPSCs were completely abolished (*Figure 1—figure supplement 1D*, control peak amplitude = 1320 ± 164 pA, TTX+4-AP peak amplitude = 15 ± 4 pA, p < 0.01). Hence, evoked EPSCs represent monosynaptic thalamocortical inputs, and evoked IPSCs represent disynaptic inputs from local inhibitory interneurons driven by thalamocortical activation.

Notably, because thalamocortical E-I ratio fails to shift toward inhibition in both V1b and V1m, depression at the thalamocortical synapses onto pyramidal neurons alone cannot account for the well-documented reduction in visual responsiveness induced by brief MD, prompting us to search

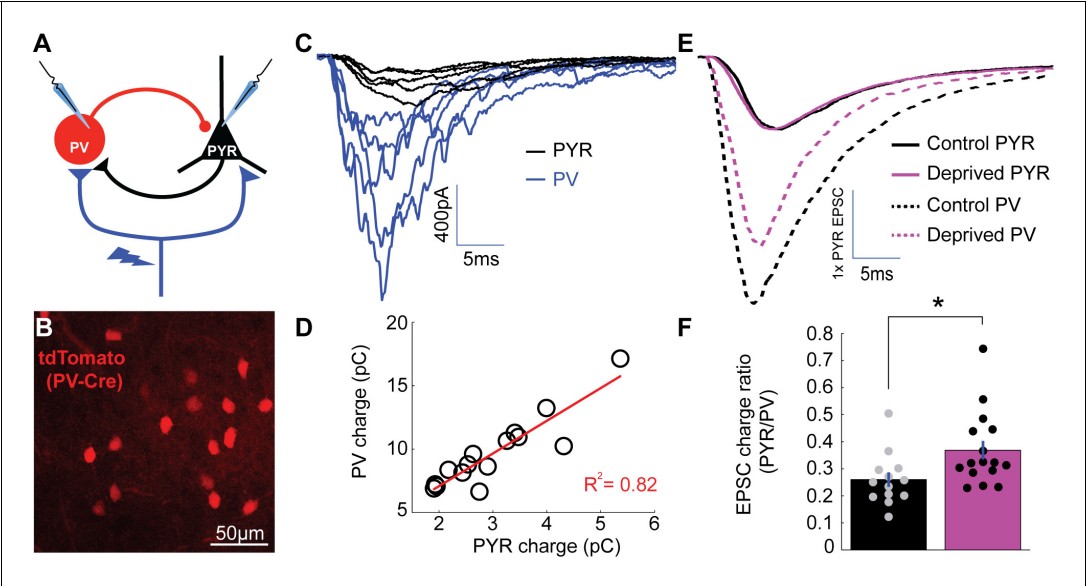

**Figure 4.** Enhanced depression at thalamocortical synapses onto PV+ interneurons following brief MD. (**A**) Whole-cell recordings were obtained from pairs of pyramidal neurons (black) and nearby PV+ interneurons (blue) within V1m while stimulating local ChR2-mCherry+ thalamocortical axons (red). (**B**) PV+ interneurons were targeted by reporter (tdTomato) expression. (**C**) Representative example traces from a pyramidal and PV pair while stimulating thalamocortical axons at a range of stimulus intensities, with pyramidal traces in black (PYR) and PV+ interneuron traces in blue (PV). Note the larger thalamocortical EPSCs in the PV+ interneuron compared to the pyramidal neuron. (**D**) Charge of thalamocortical EPSCs in the pyramidal neuron plotted against corresponding thalamocortical EPSCs in the PV+ interneuron. Linear fit plotted in red ($R^2$ = 0.82). (**E**) Averaged traces from all pairs (control in black and deprived in magenta) normalized to pyramidal EPSC peak amplitude. (**F**) Evoked thalamocortical EPSC charge ratio (PYR/PV) for control (black) and deprived (magenta) pairs (control n = 13 pairs, deprived n = 16 pairs, from 13 animals; p = 0.0103, Wilcoxon rank sum test).
DOI: https://doi.org/10.7554/eLife.38846.010

The following source data and figure supplement are available for figure 4:

**Source data 1.** Source Data for *Figure 4*.
DOI: https://doi.org/10.7554/eLife.38846.012
**Figure supplement 1.** PV+ interneurons readily fire to thalamocortical stimulation, whereas pyramidal neurons do not.
DOI: https://doi.org/10.7554/eLife.38846.011

for additional microcircuit changes within layer 4. This analysis is complicated in V1b because visual drive arises from both eyes and the ratio of this drive varies substantially between individual neurons (*Mrsic-Flogel et al., 2007*), and there is currently no viable approach in V1 slices to identify neurons and synapses driven predominantly by the deprived eye. The remainder of these experiments were therefore carried out in V1m, where the circuit is uniformly deprived.

## Brief MD induces LTD-like depression at thalamocortical synapses onto pyramidal neurons

Brief visual deprivation reduces the ability of thalamocortical synapses to evoke responses in V1 (*Heynen et al., 2003*; *Frenkel and Bear, 2004*; *Khibnik et al., 2010*) and occludes the induction of LTD at synapses onto principal neurons in layer 4 (*Crozier et al., 2007*). To determine whether brief MD induces an absolute reduction in thalamocortical postsynaptic strength, we developed a protocol that allowed us to probe quantal amplitudes at thalamocortical synapses. Specifically, we developed an optogenetically-evoked, desynchronized vesicle release paradigm to measure quantal amplitudes at labeled thalamocortical synapses onto layer 4 pyramidal neurons; note that in layer 4 of V1, the vast majority of glutamatergic principal neurons are pyramidal neurons with a thin apical dendrite that extends into layer 2/3 (*Peters and Kara, 1985*; *Maffei et al., 2004*). We obtained whole-cell recordings from layer 4 pyramidal neurons in the presence of tetrodotoxin (TTX) to block action potentials, 4-aminopyridine (4-AP) to enhance the excitability of presynaptic terminals (*Petreanu et al., 2009*), and (2R)-amino-5-phosphonovaleric acid (AP5) and picrotoxin (PTX) to isolate AMPA-mediated mEPSCs, then illuminated a 50 μm spot encompassing labeled axons near the

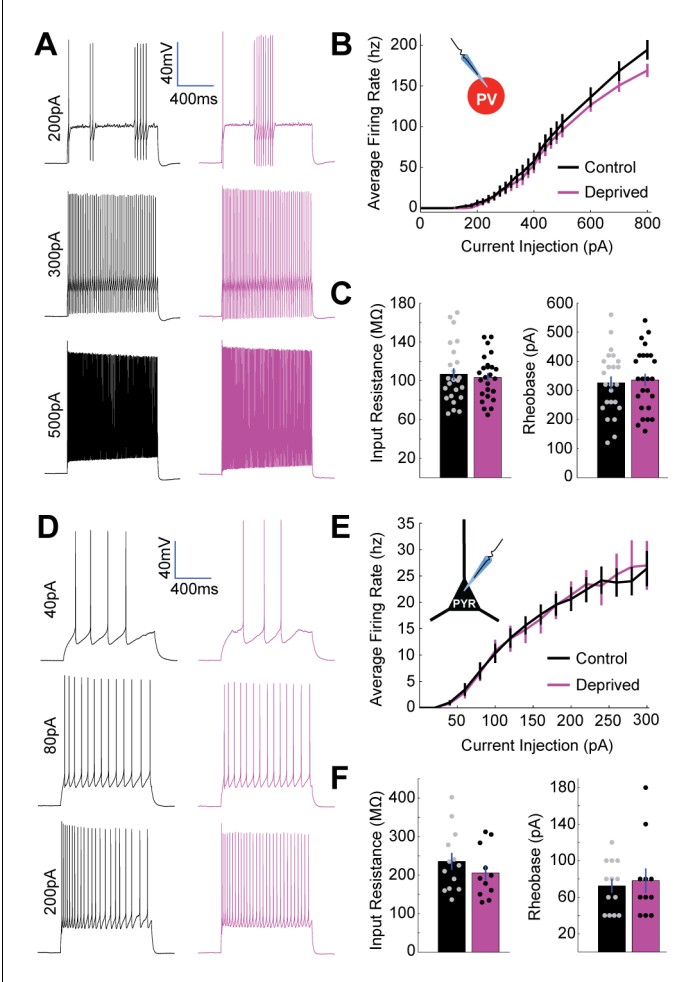

**Figure 5.** No change in intrinsic excitability of layer 4 PV+ interneurons or pyramidal neurons following brief MD. (**A**) Representative example traces from a control PV+ interneuron (black) and a deprived PV+ interneuron (magenta) in V1m showing firing responses to 1 s long current injections of 200 pA (top), 300 pA (middle), and 500 pA (bottom). (**B**) PV+ interneuron firing rate versus current injection (FI) plotted for all neurons in control (black, n = 24) or deprived (magenta, n = 25) conditions; data obtained from 10 animals. (**C**) Mean input resistance (left) and mean rheobase (right) for control (black) and deprived (magenta) PV+ interneurons. (**D**) Representative example traces from control (black) and deprived (magenta) layer 4 pyramidal neurons showing firing responses to 1 s long current injections of 40 pA (top), 80 pA (middle), and 200 pA (bottom). (**E**) Pyramidal neuron firing rate versus current injection (FI) plotted for all neurons in control (black, n = 13) or deprived (magenta, n = 11) conditions; data obtained from seven animals. (**F**) Mean input resistance (left) and mean rheobase (right) for control (black) and deprived (magenta) pyramidal neurons.

DOI: https://doi.org/10.7554/eLife.38846.013

The following source data is available for figure 5:

**Source data 1.** Source Data for *Figure 5*.

DOI: https://doi.org/10.7554/eLife.38846.014

soma of the recorded neuron. Under these conditions, long (1–5 s) pulses of blue laser light elevated mEPSC frequency to (on average) 485 ± 48% of baseline (*Figure 2A,B*, *Figure 2—figure supplement 1A*, p < 0.0001). At moderate stimulus intensities, this allowed us to measure evoked quantal events from thalamocortical synapses with minimal contamination from spontaneous mEPSCs. Examination of the amplitude distributions of evoked events revealed a similar distribution to spontaneous events, with no multimodal peaks indicative of contamination by multiquantal events (*Neubig et al., 2003*); *Figure 2—figure supplement 1B*). The mean amplitude and kinetics of

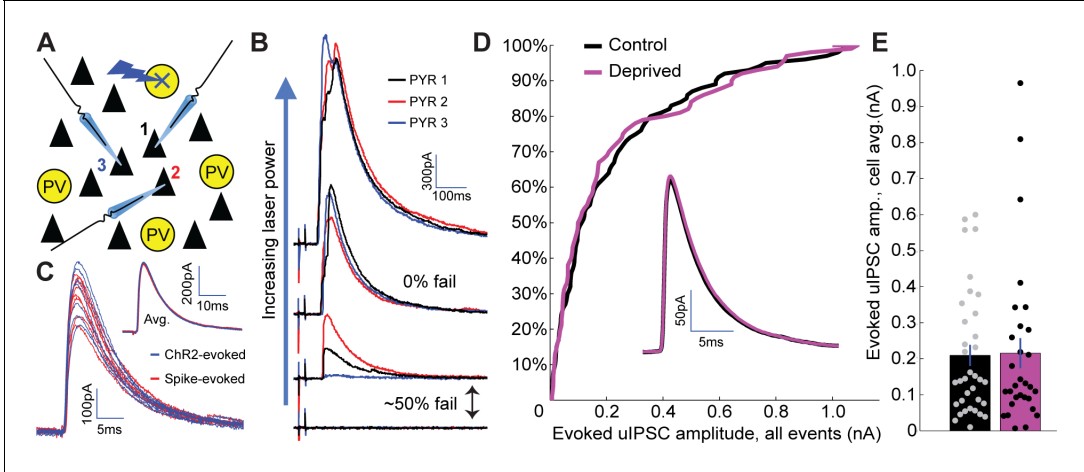

**Figure 6.** No change in PV+ interneuron to pyramidal neuron inhibitory synaptic strength following brief MD. (**A**) Recording/stimulating configuration. While recording from multiple pyramidal neurons within V1m, nearby PV+ interneurons expressing ChR2-YFP were targeted serially with focal laser stimulation. (**B**) For each targeted PV+ interneuron, laser stimulation strength was initially very low and was gradually increased until one or more recorded pyramidal neurons exhibited ~50% rate of failure in evoked IPSCs. (**C**) Example IPSCs evoked using optogenetic stimulation (blue) or by patching the same PV+ interneuron and evoking individual spikes with current injection (red). Inset: average traces from each method overlaid. (**D**) Cumulative probability distribution for all putative unitary IPSC amplitudes (includes multiple inputs per pyramidal neuron) is not significantly different between control and deprived conditions (control n = 68, deprived n = 59, from 11 animals; p = 0.90, two-sample Kolmogorov-Smirnov test). Inset: average IPSC waveforms (cell-averages) are indistinguishable between control and deprived pyramidal neurons. (**E**) Quantification of cell-averaged putative unitary IPSC amplitudes shows no significant difference between control and deprived pyramidal neurons (control amplitude = 209 ± 29 pA n = 35 neurons, deprived amplitude = 215 ± 42 pA n = 30 neurons, p = 0.73, Wilcoxon rank sum test).

DOI: https://doi.org/10.7554/eLife.38846.015

The following source data is available for figure 6:

**Source data 1.** Source Data for *Figure 6*.

DOI: https://doi.org/10.7554/eLife.38846.016

evoked thalamocortical quantal events were similar to spontaneous quantal events, as expected (*Gil et al., 1999*), *Figure 2—figure supplement 1C*). Brief MD induced a modest but significant reduction in evoked thalamocortical quantal amplitude (*Figure 2C*, control = 10.7 pA ± 0.35 pA, deprived = 9.7 pA ± 0.20 pA, p < 0.05), without any impact on input resistance, resting membrane potential (*Figure 2D*), waveform kinetics (*Figure 2E*), or spontaneous mEPSC amplitudes (which are enriched for intracortical excitatory synapses, *Figure 2—figure supplement 1D*). This reduction in quantal amplitude following brief MD is consistent with the induction of a postsynaptic depression selectively at thalamocortical synapses onto layer 4 pyramidal neurons.

If this reduction in thalamocortical quantal amplitude reflects the induction of LTD, then further LTD induction should be occluded. To test this, we adapted a low frequency LTD induction paradigm (*Crozier et al., 2007*) for optogenetic stimulation, which has the advantage over the standard approach of extracellular electrical stimulation (e.g. *Crozier et al., 2007*) of selectively activating only thalamocortical axons. We adjusted the laser intensity to evoke stable thalamocortical EPSCs of 200–800 pA; note that average baseline amplitudes were similar between conditions (*Figure 3E,F*). By pairing 1 Hz activation of thalamocortical axons with brief postsynaptic depolarization to −50 mV (in voltage clamp) for 10 min, we reliably induced sustained depression of evoked thalamocortical EPSCs onto control neurons (*Figure 3A,B,E*, 64.4 ± 6.7% of baseline amplitude). In contrast, deprived neurons exhibited significantly less depression following LTD induction (*Figure 3B,F,G*, 89.3 ± 9.7% of baseline amplitude, p < 0.05). There were no changes in passive properties following LTD induction in either condition (*Figure 3C,D*). Taken together with the change in thalamocortical quantal amplitude, this strongly suggests that brief MD induces a postsynaptic LTD-like depression of thalamocortical synapses onto layer 4 pyramidal neurons. Coupled with our previous result that thalamocortical-evoked E-I ratio is shifted toward excitation following brief MD (*Figure 1G*), this implies that there is a concurrent and even greater weakening of thalamocortical-recruited feedforward inhibition onto pyramidal neurons.

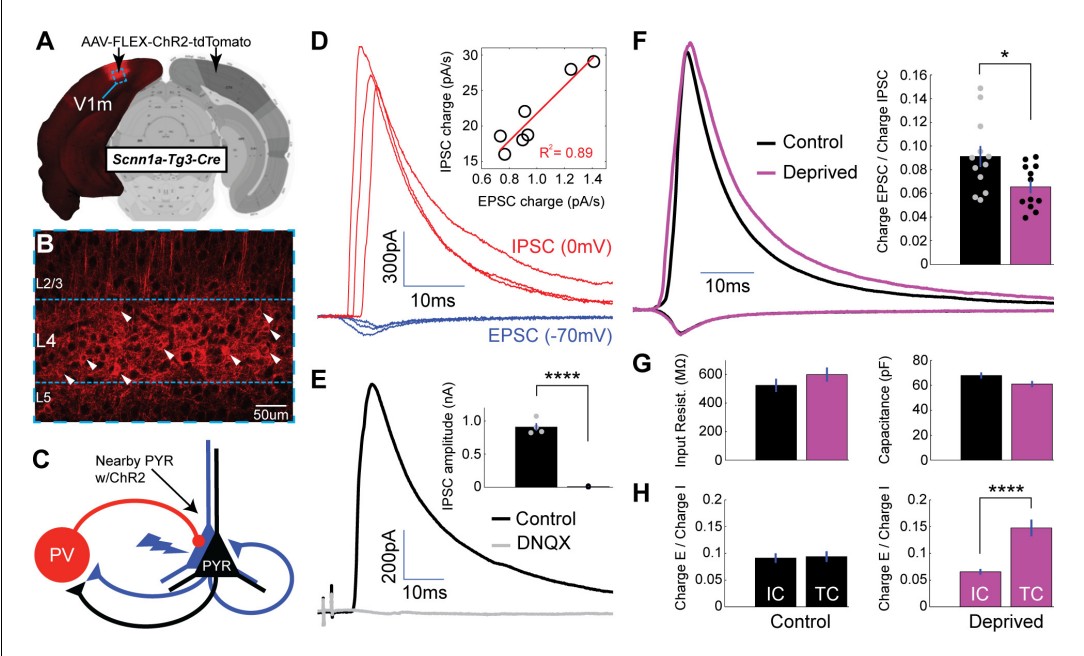

**Figure 7.** Local intracortical-evoked E-I ratio shifts toward inhibition following brief MD. (**A**) Cre-dependent ChR2-tdTomato expression (red) from stereotaxic injection of AAV into V1m of Scnn1a-Tg3-Cre mice. (**B**) Confocal image from site of viral infection, with infected layer 4 pyramidal neurons marked with white arrows. (**C**) Diagram of recording configuration: whole-cell recordings were obtained from uninfected layer 4 pyramidal neurons while stimulating neighboring ChR2-tdTomato+ pyramidal neurons with brief (2 ms) pulses of 473 nm light. (**D**) Representative example paired EPSCs (red) and IPSCs (blue) recorded at experimentally verified reversal potentials (−70 mV and 0 mV) at several stimulus intensities. Inset: EPSC versus IPSC charge elicited by a range of stimulus intensities for a single neuron, with linear fit plotted in red ($R^2 = 0.89$). (**E**) Evoked IPSCs in regular ACSF (black) and following perfusion of DNQX (grey) (n = 4). (**F**) Mean EPSC charge-normalized EPSC and IPSC traces for control (black) and deprived (magenta) neurons. Right: Mean E-I charge ratio for control (black) and deprived (magenta) neurons (control n = 12, deprived n = 12, from eight animals; p = 0.022, 2-sample t-test). (**G**) Mean input resistance and capacitance for control (black) and deprived (magenta) neurons. (**H**) Left: control intracortical (IC) versus thalamocortical (TC) E-I ratios (p = 0.84, 2-way t-test). Right: deprived intracortical versus thalamocortical E-I ratios (p = $4.1 \times 10^{-5}$, 2-sample t-test). Atlas image in A) adapted from Allen Mouse Brain Atlas (*Dong, 2008*).

DOI: https://doi.org/10.7554/eLife.38846.017

The following source data is available for figure 7:

**Source data 1.** Source Data for *Figure 7*.
DOI: https://doi.org/10.7554/eLife.38846.018

## Enhanced depression at thalamocortical synapses onto PV + interneurons following brief MD

One possibility that could explain a shift in thalamocortical-evoked E-I ratio toward excitation (*Figure 1G*) is that brief MD induces depression at thalamocortical synapses onto *both* pyramidal neurons and GABAergic interneurons, but the relative decrease is *greater* onto GABAergic interneurons. The two major subtypes of GABAergic interneurons in layer 4 are parvalbumin (PV)+ and somatostatin (SST)+ interneurons (*Ji et al., 2016*; *Pfeffer et al., 2013*; *Rudy et al., 2011*). PV+ interneurons receive much stronger thalamic drive than SST+ interneurons (*Beierlein et al., 2003*; *Cruikshank et al., 2010*; *Ji et al., 2016*; *Urban-Ciecko and Barth, 2016*; *Yavorska and Wehr, 2016*), suggesting that they mediate the vast majority of thalamocortical-evoked feedforward inhibition. To target PV+ interneurons, we crossed mice carrying PV-Cre (*Hippenmeyer et al., 2005*) and Rosa26-STOP-tdTomato (Ai9, *Madisen et al., 2010*) alleles, such that progeny express tdTomato in PV+ interneurons (*Figure 4B*). To verify that layer 4 PV+ interneurons in V1 fire in response to thalamocortical activation in the range of stimulus intensities used to assess E-I ratio (*Figure 1E*), we recorded in current clamp from tdTomato-expressing PV+ interneurons and nearby pyramidal neurons in layer 4 while stimulating thalamocortical axons. Indeed, PV+ interneurons fired readily even to modest thalamocortical stimulation and with a much lower threshold than pyramidal neurons

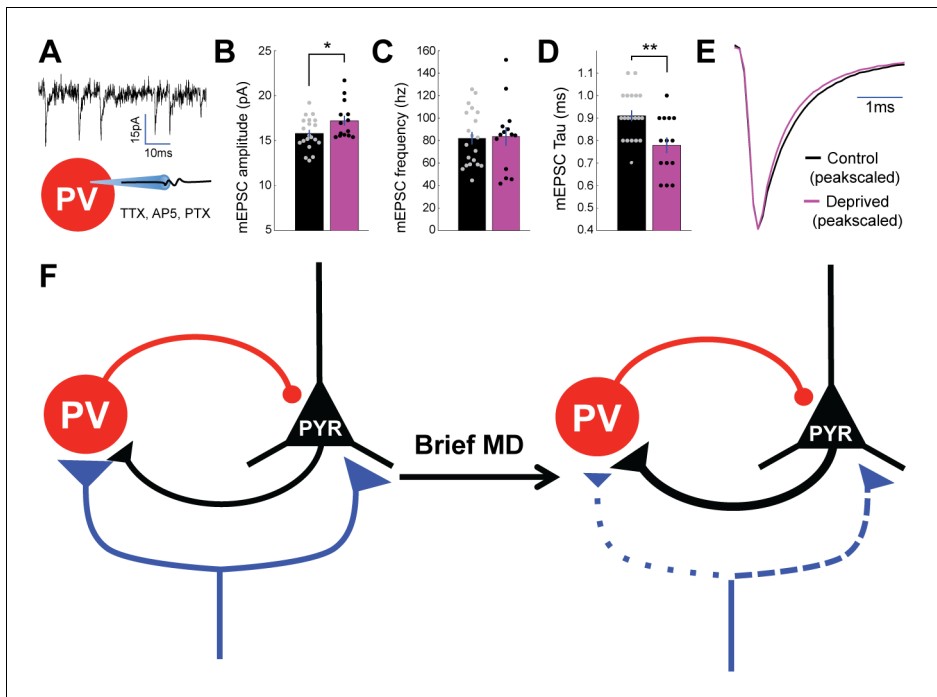

**Figure 8.** Spontaneous mEPSC amplitude onto PV+ interneurons is enhanced by brief MD. (**A**) Whole cell recordings were performed in V1m on PV+ interneurons in TTX, AP5, and PTX to isolate spontaneous mEPSCs. (**B**) Quantification of mEPSCs shows a significant potentiation in amplitude following brief MD (p = 0.044, Wilcoxon rank sum test), (**C**) no change in event frequency (p = 0.86, 2-sample t-test), and (**D**) a significant decrease in event decay constant (p = 0.003, 2-sample t-test. Control n = 20, deprived n = 14, from 10 animals). (**E**) Average mEPSC waveforms (left) and peak-scaled average mEPSC waveforms (right) for interneurons from control (black) and deprived (magenta) hemispheres. (**F**) Scheme summarizing changes within layer 4 circuit following brief MD, with synapse size and line width corresponding to synaptic strength and dotted lines specifically indicating weaker synaptic strength. Specifically, MD leads to depression of thalamocortical strength onto both pyramidal neurons and PV+ interneurons, though this depression is greater onto PV+ interneurons. MD also strengthens excitatory connections from pyramidal neurons to PV+ interneurons.

DOI: https://doi.org/10.7554/eLife.38846.019

The following source data is available for figure 8:

**Source data 1.** Source Data for *Figure 8*.
DOI: https://doi.org/10.7554/eLife.38846.020

---

(*Figure 4—figure supplement 1*, p < 0.0001), suggesting that these neurons contribute substantially to the feedforward E-I measurements.

To test whether thalamocortical synapses onto PV+ interneurons are depressed more than thalamocortical synapses onto pyramidal neurons, we measured relative thalamocortical drive to both cell types using paired recordings in tandem with thalamocortical stimulation (*Figure 4A*). Evoked thalamocortical EPSCs onto PV+ interneurons were larger with faster kinetics than EPSCs onto pyramidal neurons as expected (*Figure 4C*; *Hull et al., 2009*; *Kloc and Maffei, 2014*), and EPSC amplitude onto the two cell types scaled linearly with stimulus intensity (*Figure 4D*). After brief MD, evoked thalamocortical EPSCs onto PV+ interneurons were significantly smaller relative to paired neighboring pyramidal neurons (*Figure 4E*), resulting in an increase in thalamocortical-evoked pyramidal/PV EPSC charge ratio (*Figure 4F*, control ratio = 0.260 ± 0.027, deprived ratio = 0.369 ± 0.034, p < 0.05). These data show that brief MD induces a greater depression of thalamocortical synapses onto layer 4 PV+ interneurons than onto neighboring pyramidal neurons; this in turn should reduce the ability of thalamocortical stimulation to recruit PV-mediated inhibition onto layer 4 pyramidal neurons and thus is predicted to contribute to the increased thalamocortical-evoked E-I ratio we observed.

## Brief MD does not affect intrinsic excitability of layer 4 neurons or inhibitory strength from PV+ interneurons to pyramidal neurons

In addition to changes at thalamocortical synapses, relative changes in intrinsic excitability of PV + interneurons and pyramidal neurons could potentially contribute to the increase in thalamocortical E-I ratio induced by brief MD. We measured intrinsic excitability of layer 4 pyramidal neurons and PV + interneurons (targeted using the reporter line as described above), by generating firing rate versus current (FI) curves under conditions where synaptic inputs were blocked (*Figure 5*). There was no significant difference in firing rate between deprived and control PV+ interneurons at any current injection (*Figure 5B*). Furthermore, there was no difference in input resistance or rheobase (*Figure 5C*). Similarly, pyramidal intrinsic excitability, input resistance, and rheobase were unaffected by brief MD (*Figure 5D–F*).

Weakening of inhibitory synaptic strength from PV+ interneurons onto layer 4 pyramidal neurons is another mechanism that could contribute to the increased E-I ratio following brief MD. Because unitary synaptic strength varied greatly at PV+ to pyramidal synapses (from 10's of pA to more than a nA, *Figure 6D*), we required a relatively high-throughput means to probe changes in strength at this synapse. To achieve this, we developed an optogenetic minimal stimulation paradigm to obtain

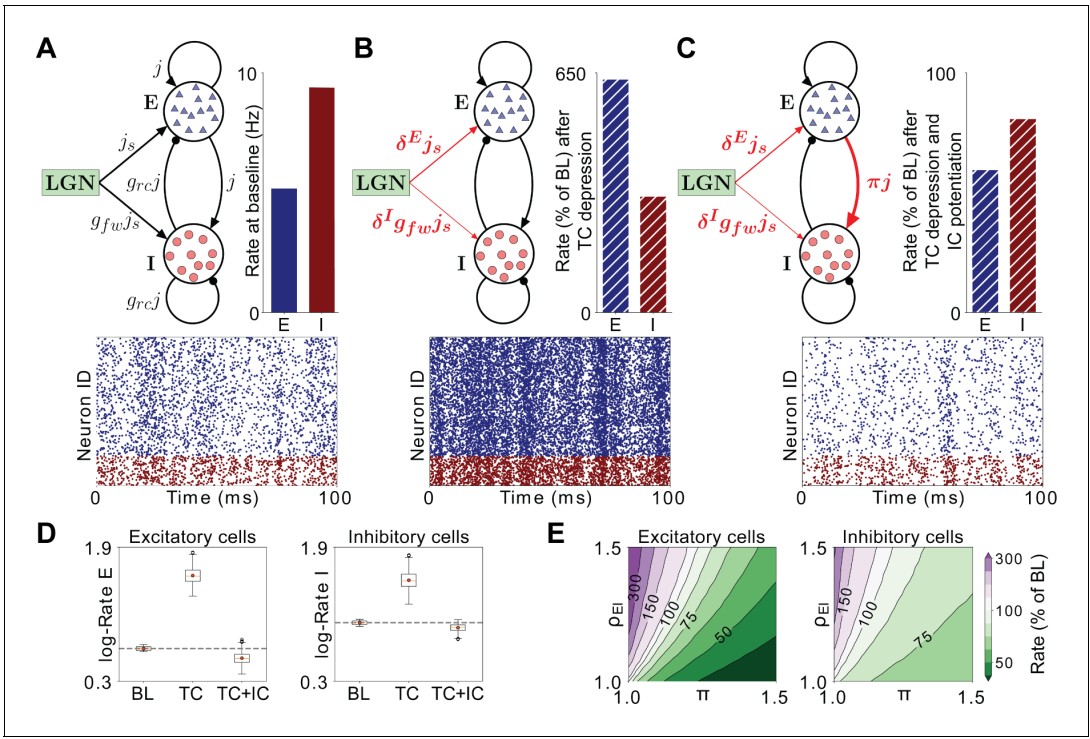

**Figure 9.** Firing rates in a model layer 4 circuit incorporating MD-induced synaptic changes in V1m. (A) Schematic of the network model under baseline conditions. All excitatory synaptic weights in the network are given by the parameter $j$, and inhibitory connections are further scaled with a factor $g_{rc}$. Excitatory neurons receive thalamic input with synaptic weight $j_s$, while the stronger thalamic input onto inhibitory neurons is captured by multiplying $j_s$ with the factor $g_{fw} > 1$. Bar plot: Firing rates of excitatory (E, red) and inhibitory (I, blue) neurons in the model at baseline (BL). Raster plot: Representative spiking activity across neurons over 100 ms at BL. (B) Schematic of the network model after implementing synaptic depression in thalamocortical (TC) projections onto excitatory neurons ($\delta^E$) and greater depression onto inhibitory neurons ($\delta^I$). Bar plot: firing rates as a percentage of BL resulting from these synaptic changes. Raster plot: representative spiking activity across neurons over 100 ms after implementing TC depression. (C) Same as B), but with the addition of intracortical (IC) potentiation ($\pi$) of recurrent excitatory drive to inhibitory neurons (TC + IC). (D) Box plots of the firing rates in 2491 implementations of the network with randomly chosen parameters, simulated at BL, after TC depression and after TC depression and IC potentiation. The boxplots show the interquartile range (IQR) of the simulated firing rates (red symbol denotes the mean, red line denotes the median), with the whiskers denoting ± 1.5 IQR. Outliers (black circles) are firing rates beyond the ± 1.5 IQR. Rates are shown on a logarithmic scale for excitatory neurons (left panel) and inhibitory neurons (right panel). (E) Combined effects of increased TC E-I ratio ($\rho_{EI}$) and potentiation of recurrent excitatory drive to inhibition ($\pi$) on firing rates of excitatory (left panel) and inhibitory neurons (right panel). Firing rates are shown as percentage of BL for all combinations of $\pi$ and $\rho_{EI}$. Green-shifted color regions represent lower rates and purple-shifted regions represent higher rates relative to BL.
DOI: https://doi.org/10.7554/eLife.38846.021

putative unitary IPSCs from many individually targeted PV+ interneurons onto nearby layer 4 pyramidal neurons. For this experiment, we crossed mice carrying PV-Cre and Rosa26-STOP-ChR2-EYFP (Ai32, *Madisen et al., 2012*) alleles, such that progeny express ChR2-EYFP in PV+ interneurons. While recording from layer 4 pyramidal neurons, nearby (within 150 μm) individual reporter-expressing PV+ interneurons were targeted with a low-intensity focal laser spot (*Figure 6A*), and laser intensity was iteratively increased until IPSCs were elicited by ~50% of stimuli (*Figure 6B*). This procedure was repeated for up to four nearby PV+ interneurons while recording from up to three layer 4 pyramidal neurons at a time, allowing us to measure many putative unitary IPSC amplitudes in each slice. Frequently, successes or failures were observed simultaneously in all patched pyramidal neurons (*Figure 6B*). Also, inhibitory connection strength varied markedly across connections, even from the same presynaptic PV+ interneuron (*Figure 6B,D*), with a small population of very strong connections (*Figure 6D*). By patching onto the same PV+ interneuron we had stimulated optogenetically, we could show that spike-evoked unitary IPSCs and optogenetically evoked putative unitary IPSCs were indistinguishable (*Figure 6C*), indicating that this approach generates reliable estimates of unitary connection strength.

Comparing optogenetically evoked putative unitary IPSC amplitude between control and deprived neurons revealed no significant difference in mean amplitude (control amplitude = 209 ± 29 pA, deprived amplitude = 215 ± 42 pA, p = 0.73), or in the distribution of IPSC amplitudes (p = 0.90), between conditions (*Figure 6D,E*). Hence, changes in unitary synaptic strength from PV+ interneurons to pyramidal neurons do not significantly contribute to the weakening of thalamocortical-recruited inhibition following brief MD. Taken together, our data suggest that the major factor that increases feedforward thalamocortical E-I ratio onto pyramidal neurons is the disproportionate reduction in thalamic drive to PV+ interneurons.

## Local intracortical-evoked E-I ratio shifts toward inhibition following brief MD

It is well established that 2–3 d of MD reduces visual responsiveness (*Heynen et al., 2003*; *Kaneko et al., 2008*), signal propagation (*Wang et al., 2011*), and firing rates (*Hengen et al., 2013*; *Hengen et al., 2016*) in V1m. Loss of visual responsiveness following brief MD has largely been ascribed to depression at thalamocortical synapses in both V1b and V1m (*Espinosa and Stryker, 2012*; *Khibnik et al., 2010*; *Smith et al., 2009*; *Wang et al., 2013*), yet here we show that the thalamocortical-evoked E-I ratio onto layer 4 pyramidal neurons is increased, rather than reduced, by brief MD in V1m. One explanation for how layer 4 excitability decreases despite this increase in thalamocortical E-I ratio is that E-I ratio is regulated differently in feedforward thalamocortical circuitry and recurrent feedback intracortical circuitry. Since intracortical excitatory synapses outnumber thalamocortical synapses by a factor of ~5 in neocortical layer 4 (*Bopp et al., 2017*) and by at least a factor of 2 onto PV+ interneurons (*Kameda et al., 2012*), a change in intracortical E-I ratio could potentially outweigh changes in thalamocortical circuitry to suppress excitability.

To measure intracortical E-I ratio in layer 4, we modified our optogenetic paradigm to allow us to label and activate a subset of layer 4 pyramidal neurons (*Figure 7A*) while recording from nearby uninfected pyramidal neurons. To achieve this, we injected a low volume of AAV-FLEX-ChR2-tdTomato into V1m of Scnn1a-Tg3-Cre mice (*Madisen et al., 2010*), resulting in a fraction of layer 4 pyramidal neurons expressing ChR2-tdTomato (*Figure 7B,C*). Activation of labeled neurons evoked monosynaptic EPSCs (from pyramid-to-pyramid synapses) and disynaptic IPSCs (from pyramid-to-GABAergic interneuron-to pyramid synapses, *Figure 7C*). These evoked currents were recorded in voltage clamp at the experimentally verified reversal potentials for inhibition and excitation. As observed for thalamocortical-evoked E-I events, EPSCs versus IPSCs scaled linearly as a function of stimulus intensities (*Figure 7D*, inset). Following perfusion of DNQX, IPSCs were abolished (*Figure 7E*, baseline IPSC amplitude = 909 ± 57 pA, DNQX IPSC amplitude = 12.2 ± 3.8 pA, p < 0.0001), indicating that ChR2-tdTomato expression was restricted to excitatory neurons. After brief MD, intracortical-evoked E-I ratios were significantly shifted toward greater inhibition in deprived versus control pyramidal neurons (*Figure 7F*, control ratio = 0.091 ± 0.009, deprived ratio = 0.066 ± 0.005, p < 0.05), with no accompanying change in passive properties (*Figure 7G*). Interestingly, intracortical and thalamocortical E-I ratios were indistinguishable under control conditions, but diverged dramatically after brief MD (*Figure 7H*, p < 0.0001, thalamocortical E-I values replotted from *Figure 3C* for comparison). This suggests that baseline E-I ratios are strongly conserved

across feedforward and feedback microcircuits in layer 4 but can be inversely regulated by brief visual deprivation.

The shift in intracortical E-I ratio toward inhibition could be produced by an increase in inhibition onto pyramidal neurons, reduced excitation between pyramidal neurons, or an increase in excitation onto GABAergic interneurons that mediate feedback inhibition. PV+ interneuron synapses onto layer 4 pyramidal neurons are unaffected by brief MD (*Figure 6*), allowing us to rule out a change at this synapse. Further, we found no significant change in the quantal amplitude of spontaneous mEPSCs (enriched for intracortical synapses) onto pyramidal neurons after brief MD (*Figure 2—figure supplement 1D*), suggesting that there is no postsynaptic depression of pyramidal-to-pyramidal synapses. Brief MD has previously been shown to potentiate synapses from pyramidal neurons onto fast-spiking (FS) interneurons in layer 4 (*Maffei et al., 2006*), so to test for postsynaptic strengthening of excitatory synapses onto PV+ FS interneurons, we targeted PV+ interneurons for whole-cell recording as described above (*Figure 4B*), and recorded spontaneous mEPSCs (*Figure 8A*). Spontaneous mEPSC amplitude was increased following brief MD (*Figure 8B*, control amplitude = 15.8 ± 0.39 pA, deprived amplitude = 17.2 ± 0.54 pA, p < 0.05), while mEPSC frequency was unaffected (*Figure 8C*). There was a decrease in mEPSC decay time (*Figure 8D,E*; control Tau = 0.910 ± 0.024 ms, deprived Tau = 0.779 ± 0.035 ms, p < 0.01), with no change in passive properties (Rin, p = 0.33). This suggests that one major factor that reduces intracortical E-I ratio following brief MD is enhanced excitation from nearby pyramidal neurons onto PV+ interneurons, either via modification to or insertion of kinetically faster synaptic AMPA receptors at these synapses. In *Figure 8F*, we summarize the changes observed at thalamocortical and intracortical synapses in layer 4 following brief MD: the net effect of these synaptic modifications is predicted to increase thalamocortical E-I ratio while reducing intracortical E-I ratio, as we observed.

## Decreased intracortical E-I ratio can account for the MD-induced drop in firing

Given that thalamocortical and intracortical E-I ratios change in opposite directions after brief MD, we asked whether the net effect of these changes was to increase or decrease local layer 4 excitability. To address this question, we built a spiking network model of a local layer 4 circuit and tested the effects of the measured synaptic changes on circuit excitability. We numerically simulated 5,000 leaky integrate-and-fire neurons (80% excitatory and 20% inhibitory) with sparse and random connectivity implemented as conductance-based synapses (Brunel, 2000). To model thalamic drive to a local network in layer 4 of V1, neurons in the network received independent excitatory spike trains with Poisson statistics with synaptic weight $j_s$. The measured strength of the thalamic drive to inhibitory neurons was stronger than the drive to excitatory pyramidal neurons (*Figure 4*), which we included through a multiplication of the overall input weight $j_s$ with a factor $g_{fw} > 1$ for the drive to inhibitory neurons. To model input from other sources – such as intracortical feedback from neurons outside the local model circuit, or from higher cortical areas – neurons also received thalamocortical-independent excitatory spike trains. Input rates were chosen such that at baseline the networks generated excitatory and inhibitory firing rates as measured experimentally in vivo (*Figure 9A*, *Hengen et al., 2013*). Thalamocortical firing rates were fixed across conditions, as mean firing rates of thalamic relay neurons in the LGN are reportedly not affected by acute lid suture (*Linden et al., 2009*). Therefore, our model contains the minimal number of parameters needed to dissect the contribution of feedforward and feedback synaptic components analyzed experimentally in the circuit.

To test the impact of the E-I changes we measured experimentally following brief MD, we first modeled the observed change in thalamocortical E-I ratio in V1m (*Figure 1G*) by implementing depression of thalamocortical strength onto excitatory neurons by the factor $\delta^E = 0.9$, and larger depression of thalamocortical strength onto inhibitory neurons by the factor, $\delta^I < \delta^E$ with $\delta^E/\delta^I \approx 1.5$. As expected, this increased the firing of both cell types due to a net reduction in inhibitory drive (*Figure 9B*). Next, we introduced the observed shift in intracortical E-I ratio (*Figure 7F*) by potentiating intracortical excitatory strength onto inhibitory neurons by the factor $\pi$, as observed here and previously (*Figure 8B*; *Maffei et al., 2006*). This additional intracortical change exerted a powerful effect that opposed the increase in thalamocortical E-I ratio, acting to decrease both excitatory and inhibitory firing rates, similar to the change seen *in vivo* following brief MD (*Figure 9C*, *Hengen et al., 2013*).

To ensure that these results were not sensitive to the specific choice of parameters, we randomly varied the remaining parameters that could not be inferred from experiments and quantified the effects of changed feedforward and feedback E-I ratios in the different resulting networks (*Figure 9D*). In particular, we simulated 2,491 networks with different weights for overall recurrent connection strength ($j$), the relative scale of recurrent inhibitory strength ($g_{rc}$) and the thalamic and background rates, all chosen from a large range (*Figure 9D*, see also Materials and methods). Applying synaptic changes as before generated consistent changes in firing rates across all networks, demonstrating a general effect of firing rate suppression that was not specific to the particular network parameters that we chose. Finally, to analyze the relative contributions of the feedforward and feedback E-I ratios and to investigate whether the suppression of firing rate in the network was specific to the experimentally measured values or a more general phenomenon, we simulated networks with combinations of increased thalamocortical E-I ratios (by varying the ratio $\rho_{EI} = \delta^E / \delta^I$) and decreased intracortical E-I ratios (by varying the depression parameter $\pi$). The model confirmed that $\pi$ has a disproportionately larger effect on network firing compared to $\rho_{EI}$ at a wide range of values (*Figure 9E*). Hence, our model suggests that the shift in the intracortical E-I ratio in favor of inhibition dominates over the shift of thalamocortical E-I ratio in favor of excitation and is the key factor driving a reduction in V1m activity after brief MD.

## Discussion

It has long been thought that LTD at thalamocortical synapses is the major or sole cause of the rapid MD-induced loss of visual responsiveness in V1 (*Bear, 2003*; *Crozier et al., 2007*; *Espinosa and Stryker, 2012*; *Khibnik et al., 2010*; *Malenka and Bear, 2004*; *Yoon et al., 2009*). Surprisingly – despite confirming the induction of LTD at thalamocortical synapses in layer 4 – we find here that the E-I ratio of feedforward thalamocortical drive onto layer 4 pyramidal neurons either remains constant (V1b) or paradoxically *increases* (V1m) after MD, indicating that brief MD does not render the thalamocortical circuit less excitable. Further analysis in V1m revealed that the increase in thalamocortical E-I ratio is driven by disproportionate depression of thalamocortical synapses onto PV+ interrneurons, thus reducing the recruitment of feedforward inhibition in layer 4. In contrast, intracortical feedback E-I ratio onto layer 4 pyramidal neurons is reduced by brief MD, and a network model of layer 4 demonstrates that the net effect of these opposing changes in feedforward and feedback drive is to suppress firing in layer 4. Thus, feedforward and feedback E-I ratio onto layer 4 pyramidal neurons can be independently regulated by visual experience, and loss of visual responsiveness in layer 4 V1m after brief MD during the critical period is due to a reconfiguration of intracortical circuits that leads to the recruitment of excess feedback inhibition.

### LTD and loss of visual responsiveness in layer 4

There is abundant evidence that brief MD during the critical period induces LTD at thalamocortical synapses onto layer 4 pyramidal neurons in both V1m (*Heynen et al., 2003*; *Wang et al., 2013*) and V1b (*Crozier et al., 2007*; *Heynen et al., 2003*). Manipulations that block this LTD also prevent the reduction in layer 4 visual evoked potentials (VEPs) elicited by stimulation of the deprived eye (*Yoon et al., 2009*), an observation that led to the conclusion that LTD of thalamocortical inputs onto layer 4 pyramidal neurons is the primary mechanism for loss of visual responsiveness in layer 4 (*Smith et al., 2009*). However, the measure of visual responsiveness (the rapid negative-going VEP) used in *Yoon et al., 2009* primarily measures direct thalamocortical-evoked synaptic responses (*Rahmat, 2009*; *Khibnik et al., 2010*), so is not expected to capture changes in intracortical circuitry or E-I ratio that might occur independently of thalamocortical LTD.

While we confirmed that brief MD induces LTD at thalamocortical synapses onto layer 4 pyramidal neurons, we used paired recordings to show that synapses onto PV+ interneurons are more strongly depressed than synapses onto nearby pyramidal neurons, suggesting that the reduced layer 4 VEP following brief MD reflects depression at both synapse types. In V1m, this in turn leads to an enhancement of thalamocortical-evoked E-I ratio onto layer 4 pyramidal neurons (and thus net disinhibition of the feedforward thalamocortical circuit) that our modeling suggests would produce an *increase* in network excitability if unopposed by other microcircuit changes. Thus, while our data are consistent with previous reports of MD-induced thalamocortical LTD and depression of VEPs (*Crozier et al., 2007*; *Heynen et al., 2003*; *Wang et al., 2013*; *Yoon et al., 2009*), they cast doubt

on the interpretation that LTD at thalamocortical synapses underlies the loss of visual responsiveness in V1 measured using extracellular spiking or intrinsic signal imaging (*Haider and McCormick, 2009*; *Hengen et al., 2013*; *Hengen et al., 2016*), both measures that are sensitive to changes in intracortical circuitry. Instead, our data strongly support an alternative explanation in which intracortical inhibition is enhanced (relative to excitation) by brief MD, and this enhancement suppresses the excitability of the layer 4 microcircuit.

An important question is whether the microcircuit mechanisms we have identified in V1m also occur in V1b. Our data showing that thalamocortical E-I ratio in V1b stays constant following brief MD despite the well-documented depression of thalamocortical synapses onto layer 4 pyramidal neurons (*Heynen et al., 2003*; *Crozier et al., 2007*) strongly suggests that – in V1b as in V1m – there must be concurrent depression of disynaptic inhibition in the feedforward circuit. The constancy of E-I ratio in V1b is reminiscent of sensory deprivation-induced plasticity in barrel cortex (S1), where whisker plucking also disproportionately reduces excitatory drive onto layer 2/3 PV+ interneurons relative to pyramidal neurons, yet E-I ratio stays constant due to concurrent potentiation of PV+ interneuron connections onto pyramidal neurons (*House et al., 2011*; *Li et al., 2014*). Both barrel cortex and V1b receive convergent sensory inputs (from different whiskers or the two eyes, respectively), so it is possible that the differential regulation of feedforward E-I ratio in V1b and V1m reflects a tuning of the local plasticity mechanisms to these distinct characteristics of the sensory input. Importantly, in neither brain area is the net excitability of the feedforward circuit reduced by brief MD, indicating that other, presumably intracortical, changes must contribute to the loss of responsiveness to the deprived eye. It is possible to measure thalamocortical E-I ratio in V1b with reasonable accuracy, because although only ~2/3 of labeled thalamocortical axons arise from the deprived eye, there is strong evidence that thalamocortical inputs from the non-deprived eye are unchanged after 2 – 3 days of MD (*Frenkel and Bear, 2004*; *Kaneko et al., 2008*), so deprived-eye changes should still be detectable at these input synapses. In contrast, how drive from the two eyes is distributed within the intracortical feedback circuitry in V1b is largely unknown and there is currently no means of isolating intracortical synaptic changes that selectively affect the deprived eye. It therefore remains possible that the intracortical plasticity mechanisms that drive loss of responsiveness within V1b differ from those we have identified within V1m.

## Independent regulation of E-I ratio at feedforward and feedback circuits within V1m

Many studies have examined the impact of sensory deprivation on PV+ fast spiking (FS) inhibition within primary sensory cortex, using a variety of protocols to elicit and measure inhibition (*Hengen et al., 2013*; *House et al., 2011*; *Kannan et al., 2016*; *Kuhlman et al., 2013*; *Ma et al., 2013*; *Maffei et al., 2006*; *Nahmani and Turrigiano, 2014*). Together, these studies suggest that changes in intracortical inhibition are highly dynamic and depend critically on the neocortical region, the length of sensory deprivation, the source of inhibition, and the neocortical layer under investigation (*Hengen et al., 2013*; *Kannan et al., 2016*; *Kuhlman et al., 2013*; *Maffei et al., 2004*, *Maffei and Turrigiano, 2008*, *Maffei et al., 2010*). However, most studies have not separated feedforward from feedback inhibition, so the degree to which they can be independently regulated by sensory experience is unknown. In part, this is due to the difficulty in isolating these two intimately related aspects of inhibition; the same PV+ FS interneurons generate both feedforward and feedback inhibition within layer 4 and layer 2/3 (*Figure 1A*), and many methods of evoking inhibition (for example electrical or visual stimulation) will elicit both together. For example, *Ma et al., 2013* found little change in visually-evoked E-I ratio in layer 4 after 3d MD, but these results are difficult to relate to ours as this study was conducted in anesthetized animals (which affects inhibitory conductance in V1, *Haider et al., 2013*), and could not separate out the relative contribution of feedforward and feedback inhibition. Here we used an optogenetic approach to isolate and independently measure these two aspects of inhibition, and we found that in V1m these two distinct inhibitory microcircuit motifs – one mediated by thalamocortical drive and the other mediated by recurrent pyramidal drive onto the same postsynaptic PV+ interneurons – change in opposite directions following brief MD.

Although PV+ FS synapses onto layer 4 pyramidal neurons are known to be plastic (*Lefort et al., 2013*; *Maffei et al., 2004*; *Maffei et al., 2006*; *Nahmani and Turrigiano, 2014*), changes at this synapse cannot account for either MD-induced increased feedforward or decreased feedback inhibition onto layer 4 pyramidal neurons, as we observed no change in either PV+ interneuron to pyramidal

synaptic strength or in intrinsic excitability of PV+ interneurons. We probed putative unitary PV+ to pyramidal connections using ChR2 to activate individual labeled PV+ interneurons, a relatively high-throughput approach that allowed us to sample a large number of connections, and found that the distribution of synaptic weights was unaffected by brief MD. This result contrasts with a previous study that used paired recordings to demonstrate enhanced unitary connection strength at this synapse after MD from postnatal days 18–21 (*Maffei et al., 2006*). This discrepancy may reflect differences in the timing of MD, as postnatal day 18 is at the transition between the pre-critical period and the critical period proper (which opens around postnatal day 21); alternatively, it may reflect differences in sampling of the FS population, as *Maffei et al., 2006* used firing properties to identify FS cells in rats, while here we used a mouse line with abundant labeling of PV+ FS cells in layer 4.

Independent regulation of feedforward and feedback inhibition requires either that the sources of inhibition differ, or that the same source of inhibition is differentially recruited by feedforward and feedback excitatory drive. Our data support the latter model, as brief MD induces strong depression of thalamocortical synapses onto PV+ interneurons, while at the same time spontaneous mEPSCs (which largely arise from intracortical sources) onto PV+ cells are potentiated. However, it remains possible that changes in the recruitment of additional sources of inhibition contribute to the reduced intracortical E-I ratio. Under basal conditions, both feedforward and feedback inhibition in layer 4 are largely mediated by PV+ interneurons; although SST+ interneurons (the other major GABAergic cell type in layer 4) do inhibit pyramidal neurons, they receive relatively weak thalamocortical drive (*Beierlein et al., 2003*; *Cruikshank et al., 2010*; *Ji et al., 2016*) and provide much stronger inhibition onto PV+ FS interneurons than onto layer 4 pyramidal neurons (*Xu et al., 2013*), and so they are thought to primarily *disinhibit* pyramidal neurons in layer 4. Plasticity of neocortical SST+ interneurons is poorly explored and it is unknown whether sensory deprivation can modify the strength of their output synapses (*Scheyltjens and Arckens, 2016*; *Urban-Ciecko and Barth, 2016*), but reduced SST inhibition onto PV+ interneurons and/or increased SST inhibition onto pyramidal neurons are intriguing possible mechanisms that could contribute to the enhanced intracortical inhibition following brief MD. For simplicity and consistency with our data as well as known plasticity mechanisms, we modeled the change in intracortical E-I ratio as a change in excitatory drive from pyramidal neurons to FS interneurons (*Figure 9*). In this model, the shift toward enhanced intracortical inhibition outweighs the effects of reduced thalamocortical inhibition over a wide range of network parameters; this is in large part because the vast majority of input to both PV+ interneurons and pyramidal neurons arises from recurrent excitatory synapses, so that even small changes to net intracortical excitatory drive to inhibitory neurons can have a powerful impact on layer 4 firing.

## Mechanisms of E-I regulation

Eyelid closure during MD leads to a degradation of retinal input to dLGN and the decorrelation of firing of dLGN relay neurons (*Linden et al., 2009*), and this decorrelation likely drives LTD by reducing the ability of thalamocortical afferents to effectively cooperate to depolarize postsynaptic pyramidal neurons (*Smith et al., 2009*). LTD has not previously been reported at thalamocortical synapses onto PV+ interneurons. LTD has been described at a number of other hippocampal and neocortical excitatory synapses onto PV+ interneurons (*Lu et al., 2007*; *McMahon and Kauer, 1997*; *Nissen et al., 2010*), but the pre-post activity patterns and signaling pathways required for its induction are variable and depend on circuit and cell type (*Gibson et al., 2008*; *Lu et al., 2007*). Our data suggest that the same MD-induced decorrelation of thalamocortical input that drives LTD at thalamocortical synapses onto pyramidal neurons is even more effective at driving LTD at synapses onto PV+ interneurons. This imbalance in LTD induction results in an enhanced feedforward excitability, which may serve to compensate for degraded sensory input by boosting the sensitivity of the system to the input that remains (*Hennequin et al., 2017*). An interesting question is whether this reconfiguration of the feedforward network is solely an early adaptive mechanism that eventually reverses, or whether it persists even in the face of prolonged sensory deprivation. The cellular plasticity mechanisms that enhance feedback inhibition (relative to excitation) within the layer 4 recurrent circuit are less clear, although an increase in pyramidal to PV+ synaptic strength is likely an important contributor. Interestingly, unlike the change in feedforward inhibition in V1m (*Figure 1G*), the change in feedback inhibitory charge is primarily driven by a prolongation of inhibitory current rather than a change in peak current (*Figure 7F*). This suggests that the recruitment of inhibitory neurons by intracortical drive is prolonged after brief MD, presumably due to increased net excitation onto

PV+ interneurons. An intriguing question is whether this plasticity in the feedback intracortical circuit is driven by preceding changes in feedforward drive. For instance, our modeling predicts that feedforward changes alone will enhance firing of both pyramidal neurons and PV+ interneurons and enhance synchrony (*Figure 9B*), which could be sufficient to drive plasticity at intracortical synapses onto inhibitory neurons (*Caporale and Dan, 2008*; *Hennequin et al., 2017*).

What purpose might it serve to boost the gain of incoming sensory drive, only to throttle it down again by suppressing recurrent excitability? One possibility is that these combined changes serve to enhance the signal-to-noise ratio within layer 4. The increased feedforward E-I ratio should boost the ability of thalamic inputs to drive pyramidal neurons, thus offsetting to some extent the impact of degraded retinal drive. Concurrently, the prolongation of recurrent inhibition is expected to prevent these signals from being amplified and spreading out in time and space. Thus, these changes together may constrain overall activity in layer 4 to prevent the amplification of noise, while still allowing the temporal pattern of thalamocortical input to influence layer 4 pyramidal spike patterns.

## Materials and methods

### Key resources table

| Reagent type (species) or resource | Designation | Source or reference | Identifiers | Additional information |
|---|---|---|---|---|
| Strain, strain background (Rattus norvegicus) | Long-Evans Rat | Charles River Labs | Charles River 006; RRID: RGD_2308852 | |
| Strain, strain background (Mus musculus) | WT Mouse, C57BL/6J | Jackson Labs | RRID: IMSR_JAX:000664 | |
| Strain, strain background (Mus musculus) | PV-Cre Mouse, C57BL/6J | Jackson Labs; *Hippenmeyer et al., 2005* | B6;129P2-$Pvalb^{tm1(cre)Arbr}$/J; RRID:IMSR_JAX: 008069 | |
| Strain, strain background (Mus musculus) | Ai9 Mouse, C57BL/6J | Jackson Labs; *Madisen et al., 2010* | B6.Cg-$Gt(ROSA)$ $26Sor^{tm9(CAG-tdTomato)Hze}$/J; RRID:IMSR_JAX: 007909 | |
| Strain, strain background (Mus musculus) | Ai32 Mouse, C57BL/6J | Jackson Labs; *Madisen et al., 2012* | B6;129S-$Gt(ROSA)$ $26Sor^{tm32(CAG-COP4*H134R/EYFP)Hze}$/J; RRID:IMSR_JAX: 012569 | |
| Strain, strain background (Mus musculus) | Scnn1a-Tg3-Cre Mouse, C57BL/6J | Jackson Labs; *Madisen et al., 2010* | B6;C3-Tg (Scnn1a-cre)3Aibs/J; RRID:IMSR_JAX: 009613 | |
| Recombinant DNA reagent | AAV-ChR2-mCherry | UPenn Vector Core | Penn ID: AV-9–20938M; Addgene: 100054-AAV9 | |
| Recombinant DNA reagent | AAV-FLEX-ChR2-td Tomato | UPenn Vector Core | Penn ID: AV-9–18917P; Addgene: 18917-AAV9 | |

### Overview

Experiments were performed on Long-Evans rats (*Figure 1—figure supplement 1*, *Figure 2*, and *Figure 3*), WT C57BL/6J mice, (*Figure 1*), or various genetic mouse lines in C57BL/6J backgrounds (*Figure 4*, *Figure 5*, *Figure 6*, *Figure 7* and *Figure 8*, see Key resources table for details of mouse strains). For all experiments, both males and females between postnatal day (P) 25 and 28 were used for slice physiology. Litters were housed on a 12:12 light cycle with the dam (rats) or with a

mating pair (mice) with free access to food and water. Pups were weaned and moved to a new cage with 2–4 littermates at P21. Animals were provided additional bedding material, plastic huts, and wooden gnawing blocks for enrichment. All animals were housed, cared for, surgerized, and sacrificed in accordance with Brandeis IBC and IACAUC protocols. For most experiments, deprived data were obtained from V1m contralateral to the deprived eye, and control data obtained from ipsilateral V1m from the same animals; for experiments in V1b, data were obtained from deprived versus non-deprived littermates. Although it was recently shown that the ipsilateral eye can indirectly influence neurons in V1m via callosal projections, MD does not affect the strength of the ipsilateral projection to the non-deprived hemisphere (*Griffen et al., 2017*), indicating that (as our previous data has suggested, *Desai et al., 2002*; *Maffei et al., 2004*; *Maffei et al., 2006*; *Maffei and Turrigiano, 2008*; *Lambo and Turrigiano, 2013*; *Hengen et al., 2016*) the ipsilateral hemisphere is a valid control for the effects of MD on V1m. The number of animals and neurons for each experiment is given in the figure legends, but experiments were performed on a minimum of 6 animals each. Individual data points in figures represent number of neurons.

## Virus injections into dLGN and V1m

Viral injections were performed between P12-P16 using stereotaxic surgery under ketamine/xylazine/acepromazine anesthesia. Dorsal LGN or V1m layer 4 was targeted bilaterally using stereotaxic coordinates after adjusting for the lambda-bregma distance for age. A glass micropipette pulled to a fine point delivered 500 nL (rat dLGN), 200 nL (mouse dLGN), or 75 nL (mouse V1m) of virus-containing solution at the targeted depth.

## Lid sutures and ex vivo acute brain slice preparation

Lid sutures were performed between P22 and P23. Animals were anesthetized with ketamine/xylazine/acepromazine, one eye was chosen randomly for lid suture, 1–2 mm of the lower part of each eyelid was trimmed, and eyelids were sutured closed with 3 6–0 nylon or silk sutures. Sutures were checked each day and animals were only used if sutures remained intact. For brain slice preparation, animals between P25 and P27 (2 – 3 days after lid suture) were anesthetized with isoflurane, and coronal brain slices (300 µm) containing V1 were obtained from the control and deprived hemispheres of each animal. After slicing in carbogenated (95% $O_2$, 5% $CO_2$) standard ACSF (in mM: 126 NaCl, 25 $NaHCO_3$, 3 KCl, 2 $CaCl_2$, 2 $MgSO_4$, 1 $NaH_2PO_4$, 0.5 Na-Ascorbate, osmolarity adjusted to 315 mOsm with dextrose, pH 7.35), slices were immediately transferred to a warm (34°C) chamber filled with a continuously carbogenated 'protective recovery' (*Ting et al., 2014*) choline-based solution (in mM: 110 Choline-Cl, 25 $NaHCO_3$, 11.6 Na-Ascorbate, 7 $MgCl_2$, 3.1 Na-Pyruvate, 2.5 KCl, 1.25 $NaH_2PO_4$, and 0.5 $CaCl_2$, osmolarity 315 mOsm, pH 7.35) for 10 min, then transferred back to warm (34°C) carbogenated standard ACSF and incubated another 20 min. Slices were used for electrophysiology between 1 – 6 hr post-slicing.

## Electrophysiology

V1m was identified in acute slices using the shape and morphology of the white matter and hippocampal formation as a reference. Pyramidal neurons were visually targeted and identified by the presence of an apical dendrite and teardrop shaped soma, and morphology was confirmed by *post hoc* reconstruction of biocytin fills. PV+ interneurons were identified by reporter expression driven by the PV-Cre allele. Borosilicate glass recording pipettes were pulled using a Sutter P-97 micropipette puller, with acceptable tip resistances ranging from 2 to 5 MΩ. For *Figure 2*, *Figure 3*, *Figure 5* and *Figure 8*, K+ Gluconate-based internal recording solution was modified from *Lambo and Turrigiano, 2013*, and contained (in mM) 100 K-gluconate, 10 KCl, 10 HEPES, 5.37 Biocytin, 10 Na-Phosphocreatine, 4 Mg-ATP, and 0.3 Na-GTP, with sucrose added to bring osmolarity to 295 mOsm and KOH added to bring pH to 7.35. For *Figure 1*, *Figure 4*, *Figure 6* and *Figure 7*, Cs+ Methanesulfonate-based internal recording solution was modified from *Xue et al., 2014*, and contained (in mM) 115 Cs-Methanesulfonate, 10 HEPES, 10 BAPTA•4Cs, 5.37 Biocytin, 2 QX-314 Cl, 1.5 $MgCl_2$, 1 EGTA, 10 $Na_2$-Phosphocreatine, 4 ATP-Mg, and 0.3 GTP-Na, with sucrose added to bring osmolarity to 295 mOsm, and CsOH added to bring pH to 7.35. Inclusion criteria included $V_m$, $R_{in}$, and $R_s$ cutoffs as appropriate for experiment type and internal solution; values are listed below.

All recordings were performed on submerged slices, continuously perfused with carbogenated 34°C recording solution. Neurons were visualized on an Olympus BX51WI upright epifluorescence microscope using a 10x air (0.13 numerical aperture) and 40x water-immersion objective (0.8 numerical aperture) with infrared-differential interference contrast optics and an infrared CCD camera. Up to three neurons were simultaneously patched using pipettes filled with internal as described above. Data were low-pass filtered at 6 kHz and acquired at 10 kHz (except for mEPSC data in *Figure 2*, which were low-pass filtered at 2.6 kHz and acquired at 5 kHz) with Multiclamp 700B amplifiers and CV-7B headstages (Molecular Devices, Sunnyvale CA). Data were acquired using an in-house program written in Igor Pro (Wavemetrics, Lake Oswego OR), and all post-hoc data analysis was performed using in-house scripts written in MATLAB (Mathworks, Natick MA). For optogenetic photostimulation, 473 nm blue light was emitted from a 50 mW DPSS laser, which was gated by a LS2 2 mm uni-stable shutter (Vincent Associates, Rochester NY), and light intensity was controlled with a NDM2 filter wheel (Thor Labs, Newton NJ). Laser light was routed through the optical path of the microscope and focused to a ~ 50 μm diameter spot through the 40x objective.

## mEPSC recordings

For spontaneous mEPSC and evoked thalamocortical mEPSC recordings, layer 4 pyramidal neurons were voltage clamped to −70 mV in standard ACSF containing a drug cocktail of TTX (0.2 μM), APV (50 μM), picrotoxin (25 μM), and 4-AP (100 μM) with a K+ Gluconate based internal solution. 20 s traces were obtained under high gain (10-50x). Baseline traces were first obtained (for spontaneous mEPSCs) and then traces were obtained while stimulating continuously with 473 nm laser light for 2 – 5 s. Stimulus intensity was chosen such that evoked frequency was well above spontaneous frequency, but individual events were still clearly visible. Event inclusion criteria included amplitudes greater than 5 pA and rise times less than 3 ms. Neurons were excluded from analysis if $R_s$ > 20 MΩ or $V_m$ > −50 mV. Quantal data were analyzed in a fully automated manner (except for frequency data in *Figure 1—figure supplement 1C*, which involved manual analysis to accurately detect overlapping events) using in-house scripts written in MATLAB (Mathworks, Natick MA).

## LTD induction

To induce thalamocortical LTD, a low frequency stimulation (LFS) LTD induction paradigm was adapted from *Crozier et al., 2007* for use with optogenetic stimulation of thalamocortical afferents. Whole cell voltage clamp recordings were obtained from up to three layer 4 pyramidal neurons simultaneously in V1m in standard ACSF with a K+ Gluconate based internal solution, and neurons were held at −70 mV. Laser stimulus strength was adjusted until 2 ms pulses delivered once per minute evoked thalamocortical EPSCs between 100 and 700 pA; average baseline EPSC amplitude was very similar between conditions (compare pre amplitude traces in *Figure 3E and F*). Thalamocortical EPSCs were evoked once per minute for a 10 min baseline period. LFS LTD was induced in voltage clamp by pairing 1 Hz optogenetic stimulation with a brief (100 ms) postsynaptic-step depolarization from −70 mV to −50 mV for each of 600 total pulses. Following LFS LTD induction, thalamocortical EPSCs were again evoked once per minute for a 45 min post-induction period. For quantification in *Figure 3E–G*, post-induction amplitudes represent response for each neuron averaged over the 5 – 30 min following the end of LTD induction. Data were discarded if $V_m$ > −55 mV or $R_s$ changed more than 25% during the recording.

## E-I ratio

Experimental procedure for measurement of E-I ratio was adapted from *Xue et al., 2014*. Whole cell voltage clamp recordings were obtained from up to three layer 4 pyramidal neurons simultaneously in V1m in standard ACSF with a Cs+ Methanesulfonate based internal solution. EPSCs and IPSCs were evoked optogenetically by stimulating either thalamocortical afferents (thalamocortical) or other nearby pyramidal neurons (intracortical) once per minute while alternating between the reversal potential of excitation and inhibition for each stimulus. For thalamocortical E-I ratio, minimum acceptable EPSC and IPSC amplitudes were 150 pA and 600 pA, and for intracortical E-I ratio, minimum acceptable EPSC and IPSC amplitudes were 50 pA and 600 pA. Any traces that contained obvious polysynaptic activity were discarded. Recordings were excluded if $R_s$ > 25 MΩ.

## Reversal potentials

To determine the reversal potential for inhibition, we recorded from layer 4 pyramidal neurons while stimulating groups of nearby ChR2-YFP+ PV+ interneurons while holding the postsynaptic neuron at a range of potentials. The reversal potential for excitation was determined similarly, by blocking inhibition with picrotoxin while recording from layer 4 pyramidal neurons and optogenetically stimulating either thalamocortical afferents or nearby ChR2+ pyramidal neurons. The measured inhibitory and excitatory reversal potentials were close to −70 mV and 0 mV after adjusting for the liquid junction potential.

## Paired pyramidal/PV+ interneuron thalamocortical EPSCs and evoked firing

A PV+ interneuron and nearby layer 4 pyramidal neuron were patched and voltage clamped to −70 mV in standard ACSF with TTX (0.2 µM), APV (50 µM), picrotoxin (25 µM), and 4-AP (100 µM), to ensure that thalamocortical EPSCs were monosynaptic and not contaminated with IPSCs. Cs+ Methaanesulfonate based internal was used for this experiment. Thalamocortical EPSCs were evoked once per minute at a range of stimulus intensities. Recordings were excluded if $V_m > -50$ mV, $R_s > 30$ MΩ, or if there was greater than 30% difference in $R_s$ between pyramidal neuron and PV+ interneuuron within a pair. To evoke firing, neurons were recorded in current clamp with a small dc bias current to maintain resting membrane potential at −70 mV. K+ Gluconate-based internal was used for this experiment. A range of laser intensities was used for each pair.

## Intrinsic excitability measurement

Whole cell recordings were performed on layer 4 PV+ interneurons or pyramidal neurons in current clamp mode in standard ACSF with APV (50 µM), DNQX (25 µM), and picrotoxin (25 µM), using a K + Gluconate-based internal solution. A small dc bias current was injected to maintain resting membrane potential at −70 mV. Then, 1 s long current injections of increasing amplitude were delivered to generate F-I curves. Recordings were excluded if $R_s > 25$ MΩ, or $R_{in} < 100$ MΩ for pyramidal neurons or < 40 MΩ for PV+ neurons.

## Optogenetically evoked unitary IPSCs

Whole cell recordings were simultaneously performed on up to three neighboring layer 4 pyramidal neurons voltage clamped at 0 mV in standard ACSF with Cs+ Methanesulfonate based internal solution. Using epifluorescence, individual nearby ChR2-YFP+ PV+ interneurons were targeted; initial laser intensity (2 ms pulses every 30 s) was low enough to fail to evoke IPSCs, and intensity was slowly increased until ~50% of stimuli evoked an IPSC in one or more of the recorded pyramidal neurons. Other indicators that evoked IPSCs were unitary included consistent amplitude and waveform, and a tendency for all recorded pyramidal neurons to exhibit responses or failures together. Recordings were excluded if $R_s > 25$ MΩ.

## Statistical analysis

All data analysis was performed using in-house scripts written in MATLAB (Mathworks, Natick MA). For each experiment, means ±SEM derived from individual cell measurements were provided within the results section of the text, and n's (number of cells and animals), P values, and statistical tests were provided within figure captions. A 2-sample t-test was used for normally distributed data, and a Wilcoxon rank sum test was used for data that were clearly not normally distributed. Lastly, a two-sample Kolmogorov-Smirnov test was used for comparisons between distributions. Data were considered significant if p < 0.05.

## Modeling

### Modeling layer 4 network and changes due to E-I ratio

The simulated model network consisted of 5000 leaky integrate-and-fire neurons, 80% of which were excitatory and 20% inhibitory (*Brunel, 2000*). The membrane potential of each neuron was described by:

$$C\frac{dV}{dt} = -g_L(V - V_L) + I_{syn}(t)$$

where $I_{syn}(t)$ denotes the synaptic input into a neuron

$$I_{syn}(t) = -g_{exc}(t)(V - V_{exc}) - g_{inh}(t)(V - V_{inh})$$

and $g_{exc}(t)$ the total excitatory and $g_{inh}(t)$ the total inhibitory conductance. The excitatory and inhibitory reversal potentials were $V_{exc}$ and $V_{inh}$. Each cell received a total of 400 excitatory and 100 inhibitory connections from randomly chosen cells within the local circuit (probability of two cells being connected is 10%). Connections between neurons were implemented as conductance-based synapses with an exponential kernel, where upon arrival of a spike, the conductance of the postsynaptic cell increased by a fixed amount ("peak conductance") and afterwards decayed exponentially with a time constant of 5 ms. All excitatory connections within the circuit had the same peak conductance $j$ and all inhibitory connections had the same peak conductance $g_{rc}j$.

All neurons in the circuit received external excitatory drive from thalamus, modeled as 100 uncorrelated Poisson spike trains with rate $R_{LGN}$ = 8 Hz. The peak conductance for thalamic drive to excitatory cells was given by the parameter $j_s$. The strong feedforward drive to inhibitory cells, as seen in electrophysiological experiments (*Figure 4*), was included in the model as a pre-factor $g_{fw}$ to the peak conductance of thalamic inputs to inhibition. In addition to this thalamic input, both excitatory and inhibitory cells received 100 Poisson spike trains with rate $R_{BKG}$ = 14.2 Hz from an unspecified source ("background"). This could denote input from microcircuits in layer 4 farther from the one being modeled, or input from other layers. The peak conductance of these background inputs to both cell types was the same for the baseline (BL), thalamocortical (TC) and thalamocortical and intracortical (TC+IC) scenarios. All circuit and single neuron parameters are summarized in *Table 1*. This model circuit was simulated for 9 s to estimate the average population firing rate in the steady state in all three cases BL, TC and TC+IC (*Figure 9A—C*). All network simulations were performed using NEST (*Gewaltig and Diesmann, 2007*).

**Table 1.** Parameters of the neurons and network model (*Figure 9A–C,E*)

| Parameter | Sign | Value |
| --- | --- | --- |
| Total number of cells | $N$ | 5000 |
| Number of excitatory cells | $N_E$ | 4000 |
| Number of inhibitory cells | $N_I$ | 1000 |
| Connection density/probability | $\varepsilon$ | 0.1 |
| Capacitance | $C$ | 200 pF |
| Excitatory peak conductance | $j$ | 0.3 nS |
| Inhibitory peak conductance | $g_{rc}j$ | 2.4 nS |
| Leak conductance | $g_L$ | 10 nS |
| Excitatory reversal potential | $V_{exc}$ | 0 mV |
| Inhibitory reversal potential | $V_{inh}$ | −85 mV |
| Leak reversal/resting potential | $V_L$ | −70 mV |
| Firing threshold | $V_\Theta$ | −50 mV |
| Thalamic/background input weight | $j_s$ | 0.5 nS |
| Inhibitory thalamic input weight | $g_{fw}j_s$ | 0.625 nS |
| Thalamic input rate | $R_{LGN}$ | 8 Hz |
| Background input rate | $R_{BKG}$ | 14.2 Hz |
| Number of thalamic input synapses | $n_{LGN}$ | 100 |
| Number of background input synapses | $n_{BKG}$ | 100 |

DOI: https://doi.org/10.7554/eLife.38846.022

**Table 2.** Ranges of varied parameters in the Monte-Carlo simulation (*Figure 9D*).

| Parameter | Minimum | Maximum |
|---|---|---|
| Coupling scale $j$ | 0.1 nS | 0.4 nS |
| Relative inhibitory strength $g_{rc}$ | 7 | 10 |
| Thalamic input rate $R_{LGN}$ | 5 Hz | 15 Hz |
| Background input rate $R_{BKG}$ | 5 Hz | 15 Hz |

DOI: https://doi.org/10.7554/eLife.38846.023

The synaptic changes following MD were implemented by three parameters, $\delta^E$ and $\delta^I$ for depression of thalamocortical synapses onto excitatory and inhibitory neurons, respectively, and $\pi$ for potentiation of excitatory feedback onto inhibitory neurons. To simulate TC and TC+IC scenarios, the peak conductances of the baseline scenario were multiplied with these factors. For the directly measured depression of thalamocortical input synapses onto excitatory neurons to $\approx 90\%$ of the control level, we used $\delta^E = 0.9$ (*Figure 2F*). We derived the parameter for the depression of thalamocortical inputs to inhibitory cells, $\delta^I$, from the shift in the ratio of charge delivered to layer 4 pyramidal cells and PV+ cells after stimulation of thalamocortical axons (*Figure 4F*). Dividing this EPSC charge ratio in the deprived hemisphere by the EPSC charge ratio in the control hemisphere we obtained $\delta^E/\delta^I \approx 1.5$. With the measured $\delta^E = 0.9$, this implies $\delta^I = 0.6$. In a similar way, we derived the parameter $\pi$ from the change in intracortical E-I ratio (*Figure 7F*). Dividing the two ratios for control and deprived, and taking into account that synapses from PV+ cells onto layer 4 pyramidal neurons do not change following brief MD (*Figure 6*), we derived a value of $\pi = 1.5$ for the potentiation of feedback excitation onto inhibitory cells.

To test the generality of the modeling results to the choice of parameters, we used a Monte-Carlo approach. We simulated 100,000 implementations of the network with randomly chosen parameters: excitatory coupling scale $j$, the relative scaling of recurrent inhibition $g_{rc}$, and the rates of the two input sources, thalamus ($R_{LGN}$) and background ($R_{BKG}$). For each network implementation, all four parameters were chosen randomly within a certain interval (see *Table 2*). From all 100,000 implementations, we selected 2,491 networks where the firing rates in BL were matched to firing rates measured in vivo before MD (*Hengen et al., 2013*; *Hengen et al., 2016*), allowing for $\pm 10\%$ deviations around the measured firing rates. Each selected network underwent the same plastic changes corresponding to the TC and TC+IC scenarios, with the outcome summarized in *Figure 9D*.

To further investigate the interaction of the measured opposite shifts in thalamocortical and intracortical E-I ratios, we simulated a single implementation of the network on a grid of combinations of the two E-I ratios. The depression of thalamic inputs to excitatory cells was fixed to the measured value at early MD (90% of control hemisphere). This amount of depression was combined with varying amounts of depression of thalamic inputs to inhibitory cells, ranging from 90% down to 60% of their baseline value. The depression of thalamocortical synapses onto inhibitory cells was then combined with the depression of drive to excitatory cells into a single parameter $\rho_{EI} = \delta^E/\delta^I$, which accounts for the change in the thalamocortical E-I ratio. With the chosen values for $\delta^I$, $\rho_{EI}$ varied between 1 and 1.5 (y-axis in *Figure 9E*). All of the increased thalamocortical E-I ratios were then paired with decreased intracortical E-I ratios, which were implemented as potentiation of the drive from local excitatory cells onto inhibitory cells given by the parameter $\pi$ (x-axis in *Figure 9E*) and varied in the same range as $\rho_{EI}$. The firing rates for each pair ($\rho_{EI}$, $\pi$) were compared relative to the BL scenario $\rho_{EI} = \pi = 1$ (*Figure 9E*).

## Acknowledgements

Funding sources: NSF GRFP and NRSA F31 NS089170 (NJM); R37NS092635 and R01EY025613 (GGT); Max Planck Society (JG and LMAR).

## Additional information

### Funding

| Funder | Grant reference number | Author |
|---|---|---|
| National Science Foundation | NSF10604 | Nathaniel J Miska |
| National Institute of Neurological Disorders and Stroke | F31 NS089170 | Nathaniel J Miska |
| National Eye Institute | R01 EY025613 | Gina G Turrigiano |
| Max-Planck-Gesellschaft | | Julijana Gjorgjieva |
| National Institute of Neurological Disorders and Stroke | R37 NS092635 | Gina G Turrigiano |

The funders had no role in study design, data collection and interpretation, or the decision to submit the work for publication.

### Author contributions

Nathaniel J Miska, Conceptualization, Data curation, Software, Formal analysis, Funding acquisition, Investigation, Visualization, Methodology, Writing—original draft, Writing—review and editing; Leonidas MA Richter, Data curation, Software, Formal analysis, Visualization, Methodology, Writing—original draft, Writing—review and editing; Brian A Cary, Formal analysis, Investigation; Julijana Gjorgjieva, Gina G Turrigiano, Conceptualization, Resources, Supervision, Funding acquisition, Methodology, Writing—original draft, Project administration, Writing—review and editing

### Author ORCIDs

Nathaniel J Miska (iD) http://orcid.org/0000-0001-8587-4919
Julijana Gjorgjieva (iD) https://orcid.org/0000-0001-7118-4079
Gina G Turrigiano (iD) http://orcid.org/0000-0002-4476-4059

### Ethics

Animal experimentation: This study was performed in strict accordance with the recommendations in the Guide for the Care and Use of Laboratory Animals of the National Institutes of Health. All of the animals were handled according to approved Brandeis University institutional animal care and use committee (IACUC) protocols (#15005 and #18002). All surgery was performed under ketamine-xylazine-acepromazine anesthesia and included sufficient post-operative analgesia to minimize any animal suffering.

### Decision letter and Author response

Decision letter https://doi.org/10.7554/eLife.38846.026
Author response https://doi.org/10.7554/eLife.38846.027

## Additional files

### Supplementary files

• Transparent reporting form
DOI: https://doi.org/10.7554/eLife.38846.024

### Data availability

All data generated or analyzed during this study are included in the manuscript and supporting files. Individual data points are plotted over bar graphs of means +/- SEM for each figure.

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
