## [Decision Letter]

Thank you for submitting your article "Sensory experience inversely regulates feedforward and feedback excitation-inhibition ratio in rodent visual cortex" for consideration by *eLife*. Your article has been reviewed by three peer reviewers, including Julie Kauer as the Reviewing Editor, and the evaluation has been overseen by Gary Westbrook as the Senior Editor. The following individual involved in review of your submission has agreed to reveal her identity: Tara Keck (Reviewer #3).

The reviewers have discussed the reviews with one another and the Reviewing Editor has drafted this decision to help you prepare a revised submission.

Summary:

Although there is enthusiasm from the reviewers regarding the manuscript, there remain significant issues, some of which require further experiments and analysis.

Essential revisions:

First, the link to the MD binocular literature was somewhat misleading. In its present form, it is not possible to relate the manuscript's findings to binocular zone critical period plasticity. New data are required testing whether the phenomenon reported here occurs in the presence of an ipsilateral open eye. If the same effect is seen in binocular cortex, then the conclusions can stand; if not, the mechanism described in the manuscript is likely not evoked in the binocular zone, and this would significantly change the interpretation of how their results relate to existing literature; moreover, the LTD focus would be problematic.

Second, it is difficult to understand which post-MD data are reported here. Are the recordings made solely from 2-3 days post MD? Are days 2 and 3 combined? In Hengen et al. (2013) the firing rate of inhibitory neurons is decreased at 1 day MD and then rebounds, whereas firing rate of excitatory neurons decreased at 2 days MD and then recovered at 3 days MD. We'd need a strong explanation for combining the days, or preferably the data could be reported for a single MD time point. It would be ideal to determine if (a) the onset of the feedback and feedforward effects were similar, or instead if one precedes the other, or (b) determine if the intracortical increase in inhibition was transient vs. more permanent. Data on the temporal sequence of circuit changes would be very interesting and substantially increase the impact of the paper.

Finally, there was concern that Ma et al. (2013) has already showed in vivo that deprivation reduces drive from thalamus to L4, and also that inhibition contributes to the shift in ocular dominance. In the revision, it will be important to discuss the results of new experiments in the context of Ma et al.

---

## [Author Response]

Essential revisions:First, the link to the MD binocular literature was somewhat misleading. In its present form, it is not possible to relate the manuscript's findings to binocular zone critical period plasticity. New data are required testing whether the phenomenon reported here occurs in the presence of an ipsilateral open eye. If the same effect is seen in binocular cortex, then the conclusions can stand; if not, the mechanism described in the manuscript is likely not evoked in the binocular zone, and this would significantly change the interpretation of how their results relate to existing literature; moreover, the LTD focus would be problematic.

The point that we cannot assume that identical underlying mechanisms are at play V1m and V1b is well-taken. This is a complex issue that we attempt to unpack below, and to clarify in the manuscript with textual changes and additional data.

First, at a *phenomenological* level, the response in both V1m and V1b to brief monocular deprivation (MD for 1-3 days) during the classical visual system critical period has been shown to be virtually identical. In both regions of V1 there is a decrease in visual evoked potentials (VEPs) and in intrinsic signals evoked by stimulation of the deprived eye (Heynen et al., 2003; Kaneko et al., 2008); note that in both of these studies V1m and V1b were analyzed in parallel. Second, mechanistically this response depression has been attributed to LTD at excitatory synapses onto pyramidal neurons in both V1m and V1b, based on a range of approaches: (1) electrical stimulation of LGN in vivo to show depression of responses and occlusion of further LTD induction (Heynen et al., 2003); (2) occlusion of LTD induction in ex vivo slices after MD (Crozier et al., 2007); and (3) directly measuring depressed LGN EPSCs in ex vivo slices after MD (Wang et al., 2013). In some ex vivo slice experiments the brain area (V1m vs. V1b) was not specified and so presumably recordings from both areas were included (e.g. Crozier et al. 2007; Yoon et al., 2009); in one set of experiments LTD at thalamocortical synapses was directly measured in V1m after MD (Wang et al., 2013). Thus, the prior evidence for LTD induction at LGN synapses onto pyramidal neurons is strong in both brain areas, and this LTD is widely considered to be the major cause of visual response depression.

An untested prediction of this model *in both brain areas* is that feedforward thalamocortical (TC) E-I ratio should *decrease* following brief MD. In V1m we showed that instead TC E-I ratio actually *increases*. We have now added a set of experiments in V1b in which we show that TC E-I ratio is unaltered (there is a small but not significant increase) after brief MD. Thus while the magnitude of increase is smaller in V1b than V1m, in neither brain region does E-I ratio decrease as predicted if TC LTD at excitatory synapses onto pyramidal neurons is the sole or predominant mediator of loss of visual responsiveness. In light of these new data we have reordered the figures to put the TC E-I data first and have included both the V1m and V1b data together in the new Figure 1 for comparison.

Our V1b data suggest that loss of responsiveness to the deprived eye in V1b (like in V1m) is likely driven predominantly by changes in intracortical circuitry, but this is difficult to test. It is possible to measure TC E-I ratio in V1b with reasonable accuracy, because although only ~2/3 of labeled TC axons arise from the deprived eye, there is strong evidence that TC inputs from the non-deprived eye are unchanged after 2-3 days of MD (Frankel and Bear, 2004; Kaneko et al., 2008), so deprived-eye changes should still be detectible at these input synapses. In contrast, how drive from the two eyes is distributed within the intracortical feedback circuitry in V1b is largely unknown but likely varies substantially from cell to cell (e.g. Mrsic-Flogel et al., 2007), and there is currently no means of isolating intracortical synaptic changes that selectively affect the deprived eye. This was our rationale for focusing on V1m, where uniform deprivation makes it possible to derive a clear answer to the questions of how visual deprivation suppresses visual responsiveness, but we cannot say whether a similar or distinct set of plasticity mechanisms drive changes in intracortical circuitry in V1b. We now make these points clearly in a new paragraph in the Discussion subsection “LTD and loss of visual responsiveness in layer 4” (last paragraph).

Second, it is difficult to understand which post-MD data are reported here. Are the recordings made solely from 2-3 days post MD? Are days 2 and 3 combined? In Hengen et al. (2013) the firing rate of inhibitory neurons is decreased at 1 day MD and then rebounds, whereas firing rate of excitatory neurons decreased at 2 days MD and then recovered at 3 days MD. We'd need a strong explanation for combining the days, or preferably the data could be reported for a single MD time point. It would be ideal to determine if (a) the onset of the feedback and feedforward effects were similar, or instead if one precedes the other, or (b) determine if the intracortical increase in inhibition was transient vs. more permanent. Data on the temporal sequence of circuit changes would be very interesting and substantially increase the impact of the paper.

When we looked separately at our E-I data for MD 2 and MD 3 we found similar changes so we combined the data; however we do not have enough data to rule out that there are small differences in the magnitude of effects between the 2 days. We have now made clear that data from MD2 and 3 were combined (Results, first paragraph). Spreading our data over 2 days allowed us to use our genetic mouse line litters more efficiently, and we picked these time points because when we followed firing rates in individual neurons over time we found that the maximum depression in firing occurred between days 2 and 3 (see Figure 3B, Hengen et al., 2016). Further, the long literature on MD suggests that TC LTD can be detected robustly between MD 1-3, but with the maximum effect on days 2-3. While we agree that understanding the dynamics of the changes we have identified would be very interesting it would be a huge amount of additional work (each E-I dataset requires approximately 2 months to acquire given breeding times, age requirements, viral expression times, etc.) and is certainly beyond the scope of the present manuscript.

Finally, there was concern that Ma et al. (2013) has already showed in vivo that deprivation reduces drive from thalamus to L4, and also that inhibition contributes to the shift in ocular dominance. In the revision, it will be important to discuss the results of new experiments in the context of Ma et al.

The impact of brief MD on VEPs seen in Ma et al. (which we inadvertently did not cite in the original version of our manuscript) is consistent with several other in vivo publications cited (including Heynen et al., 2003 and Frenkel and Bear, 2004). The Ma et al. (2013) data on in vivo shifts in the E-I ratio are interesting but mostly become apparent after longer periods of deprivation (6d to 2 weeks); because synaptic conductances are difficult to measure accurately in vivo it may be that changes after shorter periods of time were below their ability to resolve. Two additional factors make it difficult to directly relate our results to theirs. The first is that their experiments were performed under urethane anesthesia, which is known to reduce inhibitory conductance in V1 (Haider et al., 2013), and so would be expected to impact measures of E-I ratio. The second is that they were not able to separate thalamocortical from intracortical sources of excitation and inhibition. The unique contribution of our study is that we are able to independently measure feedforward and feedback E-I ratios. We now cite Ma et al. (2013) and discuss the relevance to our data in the Discussion subsection “Independent regulation of E-I ratio at feedforward and feedback circuits within V1m” (first paragraph).